# Structure of the helicase core of Werner helicase, a key target in microsatellite instability cancers

Joseph A Newman[1], Angeline E Gavard[1], Simone Lieb[2], Madhwesh C Ravichandran[2], Katja Hauer[2], Patrick Werni[2], Leonhard Geist[2], Jark Böttcher[2], John R Engen[3], Klaus Rumpel[2], Matthias Samwer[2], Mark Petronczki[2], Opher Gileadi[1]

**Loss of WRN, a DNA repair helicase, was identified as a strong vulnerability of microsatellite instable (MSI) cancers, making WRN a promising drug target. We show that ATP binding and hydrolysis are required for genome integrity and viability of MSI cancer cells. We report a 2.2-Å crystal structure of the WRN helicase core (517–1,093), comprising the two helicase subdomains and winged helix domain but not the HRDC domain or nuclease domains. The structure highlights unusual features. First, an atypical mode of nucleotide binding that results in unusual relative positioning of the two helicase subdomains. Second, an additional β-hairpin in the second helicase subdomain and an unusual helical hairpin in the Zn²⁺ binding domain. Modelling of the WRN helicase in complex with DNA suggests roles for these features in the binding of alternative DNA structures. NMR analysis shows a weak interaction between the HRDC domain and the helicase core, indicating a possible biological role for this association. Together, this study will facilitate the structure-based development of inhibitors against WRN helicase.**

## Introduction

Werner syndrome helicase (WRN) is one of the five human members of the RecQ family of DNA helicases that unwind DNA in a 3′–5′ direction and play important roles in multiple pathways of DNA repair and maintenance of genome integrity (1). Germline defects in three of these helicases lead to syndromes with hallmarks of premature ageing and cancer predisposition: Bloom syndrome (caused by mutations in Bloom syndrome helicase [BLM]), Rothmund–Thompson syndrome (caused by mutations in RECQL4) and Werner syndrome (WS), caused by mutations in WRN helicase. To date, all WS mutations feature the introduction of premature stop codons or frame shift mutations that remove the nuclear localization signal at the C terminus of WRN (2). Individuals affected by WS display many features associated with normal human ageing, including premature greying and loss of hair, ocular cataracts, osteoporosis, atherosclerosis, and an increased risk of development of cancer, specifically thyroid cancer, melanoma, soft tissue sarcoma, and osteosarcoma. On a cellular level, cells cultured from WS patients exhibit slow growth (3), chromosome aberrations (4), genome instability, and an increased frequency of telomere shortening and loss (4) and are sensitive to various DNA damaging agents that induce inter-strand cross links (5).

WRN is a 1,432-amino acid, 162-kD polypeptide that contains a central helicase core of two domains (D1 and D2) that share homology to *Escherichia coli* RecA (residues 528–730 and 731–868, respectively), together with three additional helicase associated domains in the C terminus: a zinc-binding subdomain (869–994), a winged helix (WH) domain (956–1,064) and a helicase and RNase D C-terminal (HRDC) domain (1,140–1,239). WRN is unique among the RecQ family of helicases in containing a 3′–5′ exonuclease domain in the N terminus (residues 38–236) (6). The nuclease domain has been characterized biochemically as being inactive on single-stranded or blunt-ended double-stranded DNA but capable of cleaving single nucleotides from the 3′ end of double-stranded DNA containing 3′ recessed termini within the context of a variety of cellular DNA structures (7, 8, 9). Although WRN interacts with several proteins that participate in non-homologous end joining (10, 11), the role of WRN nuclease activity in the context of DNA repair is poorly understood. WRN-deficient cells are unable to facilitate non-homologous end joining–mediated double strand break repair (8). However, this deficiency can only be rescued by restoring both helicase and nuclease activities of WRN. Because of its interactions with various DNA repair proteins, WRN has also been implicated in several other cellular processes. Specialized functions for WRN have been found in promoting replication fork progression after DNA damage or replication fork arrest (12, 13), with WRN appearing to act in concert with the DNA2 nuclease, where it promotes degradation of reversed fork structures (14). WRN has also been implicated in base excision repair through an interaction with DNA polymerase β, stimulating its strand displacement DNA synthesis activity (15). In homologous recombination, WRN interacts with

[1]Structural Genomics Consortium, University of Oxford, Oxford, UK   [2]Boehringer Ingelheim RCV GmbH & Co KG, Vienna, Austria   [3]Department of Chemistry and Chemical Biology, Northeastern University, Boston, MA, USA

Correspondence: opher.gileadi@cmd.ox.ac.uk

RAD52 and increases its strand annealing activity; this interaction also modulates WRN helicase activity in a structure-dependent manner (16). Direct interactions of WRN with replication protein A, or shelterin complex components TRF2 and POT1, have been shown to stimulate WRN helicase activity, which is essential for DNA replication or telomere maintenance, respectively (17, 18). Consistent with WRN playing a key role in telomere maintenance, cells from WS patients display telomerase-dependent loss of telomeres from sister chromatids (4), and it has been speculated that the ability of WRN to unwind energetically stable non–B-form DNA such as G-quadruplexes may explain this phenotype.

WRN was regarded as a "guardian of genome integrity" and a tumour suppressor. However, several recent studies have identified WRN as a potent and selective vulnerability of microsatellite instability-high (MSI-H) cancers (19, 20, 21). MSI-H cancers constitute a subset of colorectal, endometrial, and gastric cancers that have deficiencies in the mismatch repair pathway and exhibit hyper-mutable state of microsatellite repeats. WRN was identified as the top dependency for MSI-H cells in two large genome-wide gene inactivation studies using either CRISPR or RNA interference, and this dependency was linked to the helicase but not the nuclease function of WRN (19, 20). These findings, together with the fact that WRN silencing in microsatellite stable (MSS) cancer and normal cells is well tolerated, suggest that WRN is a promising novel drug target for the treatment of MSI cancers. To this end, there have already been several high throughput screening efforts aimed at discovering potent and selective WRN inhibitors (22, 23, 24). And although these efforts led to compounds that induce DNA damage and apoptosis in cells (25), the mode of action and the respective binding modes are unknown, both would be of high value for the optimization into a pharmacological tool. We report the first crystal structure of the full catalytic core of WRN helicase in complex with an ADP nucleotide at 2.2-Å resolution, providing a solid basis for structure-based drug design.

## Results

### ATP hydrolysis by WRN helicase is essential for viability and genome integrity in MSI-H CRC cells

Previous studies (19, 20, 21, 26) demonstrated that WRN depletion causes pervasive DNA damage and the loss of viability selectively in MSI-H but not MSS cell lines. Mutational analyses of WRN suggested that ATP binding of the helicase domain but not enzymatic activity of the exonuclease domain is critical for the survival of MSI-H cells. It remained unknown if ATP hydrolysis by WRN helicase, and hence, ATP turnover is required for the viability of MSI-H cells and which WRN enzymatic functions are essential for maintaining genomic integrity in MSI-H cells.

To address whether ATP hydrolysis is critical for WRN function in MSI-H cells, we introduced the Walker B motif mutation E669A, predicted to abolish ATP hydrolysis, into WRN. Subsequently, we compared the E669A WRN mutant with other mutant variants containing the Walker A motif mutation K577M predicted to prevent ATP binding, and the exonuclease-dead mutation E84A (20, 27, 28).

FLAG-tagged, siRNA-resistant WRN (WRNr) transgenes that encoded wild-type or mutant variants of WRNr (Fig 1A) were stably transduced into the MSI-H colorectal cancer cell line HCT 116. Monoclonal HCT 116 lines expressing the above WRNr variants at a higher level than the endogenous level of WRN protein (Fig 1B) along with appropriate nuclear localization of transgenic proteins (Fig S1A) were chosen for further studies. Depletion of the pan-essential mitotic kinase PLK1 by siRNA abrogated cell survival in all transgenic clones (Fig 1C). Upon depletion of endogenous WRN by siRNA, HCT 116 cells harbouring the empty vector lost cell viability, whereas the expression of wild-type WRNr at both low and high levels supported cellular survival (19, 20, 21, 26), and the WRNr E84A transgene also successfully rescued the WRN depletion viability phenotype (Fig 1C). Crucially, both the Walker A mutation K577M and the Walker B mutation E669A abrogated the WRNr transgene–mediated rescue of cell viability upon depletion of endogenous WRN (Fig 1C). This result strongly suggests that both ATP-binding and ATP hydrolysis by the WRN helicase domain are essential for the survival of MSI-H cells.

Although our previous report showed that MSI-H cells fail to maintain the stability of their genome in the absence of WRN, it remained unknown whether ATP-binding and hydrolysis by the helicase domain of WRN was essential for genome integrity in MSI-H cells. To investigate this, we performed immunofluorescence analyses of the DNA damage marker γ-H2AX and scrutinized chromosome breaks in mitotic spreads. Expression of either the wild-type WRNr or the nuclease-dead WRNr E84A transgene prevented the increase in DNA damage marker (Figs 1D and E and S1B and C). In striking contrast, siRNA depletion of endogenous WRN in cells expressing either the K577M or the E669A WRNr mutant transgenes resulted in an increase in nuclear γ-H2AX levels and chromosome breaks comparable to WRN-depleted cells harbouring the empty vector (Figs 1D and E and S1B and C).

Our analyses of DNA damage markers and cell viability demonstrate that ATP-binding and ATP hydrolysis by WRN helicase, but not WRN exonuclease activity, are essential for maintaining cell viability and genomic integrity in MSI-H cells. The genome stability defects observed across the enzymatic WRN mutants strongly correlate with the observed loss of viability phenotype. This suggests that pervasive DNA damage elicited by loss of WRN protein or loss of ATP turnover by WRN helicase is responsible for the attenuated viability of MSI-H cells. Furthermore, our observations highlight the ATPase activity of WRN helicase as a therapeutic target to attack MSI-H cancers and the need for a crystal structure of the ATPase core of the WRN helicase domain. Our observations also provide pharmacodynamic markers of genome instability to track WRN ATPase inhibition in a cellular context.

### Crystal structure of the WRN ATPase core

Crystals of the WRN catalytic core were obtained using protein produced from overexpression in *E. coli* and a construct spanning residues 517–1,093 with a C-terminal hexahistidine tag and tobacco etch virus (TEV) protease–cleavage site. Crystallization was performed using sitting drop vapour diffusion and small shard-like crystals appeared between 1 and 2 mo. Crystals diffracted to 2.2-Å resolution, and the structure was solved by molecular replacement

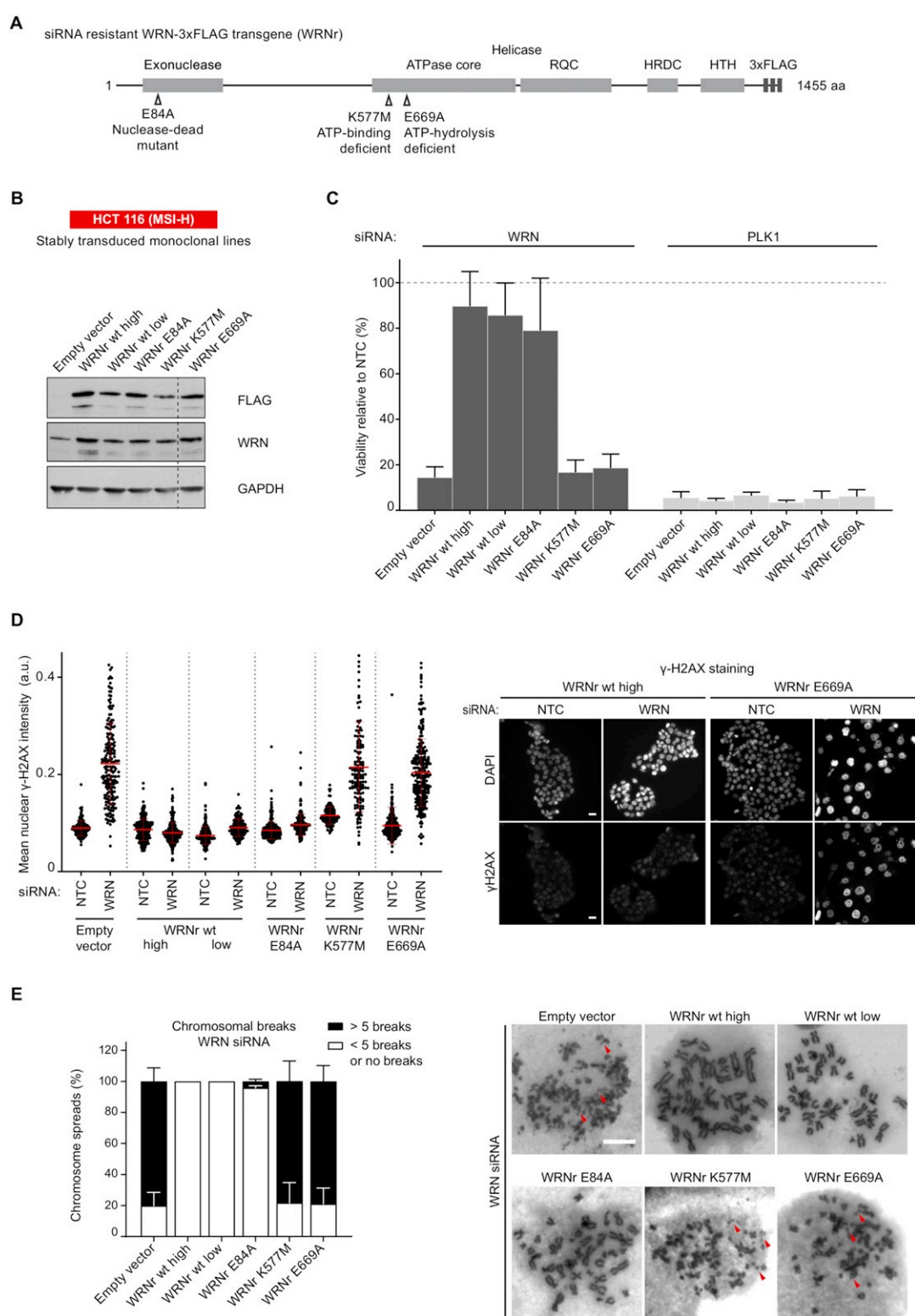

**Figure 1. ATP hydrolysis by WRN is essential for viability and genome integrity in MSI-H CRC cells.**
**(A)** Schematic representing the WRN domain structure. Location of nuclease-dead and ATPase-inactivating mutations (Walker A and B mutants) in siRNA-resistant WRN expression constructs containing a C-terminal 3xFLAG tag (WRNr) are indicated. **(B)** Monoclonal HCT 116 (MSI-H) cell clones were isolated after transduction with an empty vector control and WRNr wild-type or mutant transgenes. Immunoblotting of cell lysates with anti-FLAG and anti-WRN antibodies was used to determine the expression of the WRNr wild-type and mutant forms along with total WRN protein levels. Two WRNr wild-type clones (high and low) were selected to cover the expression range of WRNr mutant variants. **(C)** HCT 116 cells expressing WRNr transgenes were transfected with either non-targeting control or WRN siRNAs. Viability measurements were

using a domain based search strategy with the structure of RecQ1 helicase (PDBid 2WWY) as a search model for the RecA domains (29) and the WRN WH structure (PDBid 3AAF) as a search model for the WH domain (30). The crystallographic asymmetric unit contains a single molecule of WRN with no evidence for higher oligomer formation in the crystals. The model is well defined in the electron density with the exception of the first 12 residues in the N terminus, a single loop spanning residues 950–953 and the final 21 residues in the C terminus. The model has been restrained to standard bond lengths and angles with good geometry statistics (Table 1).

The overall structure of the WRN catalytic core consists of two RecA-like helicase lobes D1 and D2, each featuring a central six stranded parallel $\beta$-sheet flanked on each side by helices and loops (Fig 2A). The $Zn^{2+}$-binding subdomain features a single Zn ion tetrahedrally coordinated by four cysteine residues (C908, C935, C936, and C939) and is closely associated with the D2 domain. The WH domain extends away from the $Zn^{2+}$ and D2 domains and features a modified version of the canonical WH fold with a longer wing 2.

## Structure of the nucleotide-binding site

The nucleotide-binding site is positioned in the cleft between D1 and D2 with most contacts to the nucleotide coming from D1. The protein was crystallized in the presence of ADP, $Mg^{2+}$, aluminium chloride, and sodium fluoride, intended to produce the ATP ana-logue ADP-Aluminium fluoride, although examination of the electron density reveals only density for ADP (Fig S2); similarly, no convincing electron density could be observed for the magnesium ion. The adenine moiety on the ADP is flanked on either side by H546 and K550 and forms polar contacts to Q553, part of the conserved Q motif common to all RecQ family members (Fig 2B). The ribose makes a single contact to R857, and the phosphates are positioned directly above the N-terminal end of helix $\alpha$3 within the motif I or Walker A motif. Somewhat unexpectedly, the catalytically essential K577 does not form direct hydrogen bonds to the $\beta$-phosphate, instead forming polar contacts to motif II (Walker B motif). This is in contrast to what has been observed for other RecQ family member structures, and we expect this residue to still play an important role in WRN ATP binding, perhaps forming the contact in other con-formational states. This is also true for other residues within motif I, which generally make less direct and more water-mediated con-tacts to the phosphates than what has been observed in previous RecQ family structures (Fig S2A). On the other hand, the contacts made by residues belonging to D2 are more extensive than that observed in other RecQ structures. R857, one of two highly con-served arginines from motif VI, the so-called "arginine finger" shows a dual conformation, in which it makes contacts to both, ribose and

Table 1. Data collection and refinement statistics.

| WRN ADP | |
| --- | --- |
| Space group | P $2_1$ $2_1$ $2_1$ |
| Cell dimensions $a$, $b$, $c$ (Å) | 54.6, 90.6, 138.2 |
| Angles $\alpha$, $\beta$, $\gamma$ (°) | 90, 90, 90 |
| Wavelength (Å) | 1.08 |
| Resolution (Å) | 76.0–2.20 (2.26–2.20) |
| $R$merge | 0.13 (2.3) |
| $R$p.i.m. | 0.06 (1.13) |
| $I/\sigma I$ | 5.3 (1.0) |
| CC1/2 | 0.99 (0.50) |
| Completeness (%) | 98.8 (96.0) |
| Multiplicity | 5.5 (5.5) |
| No. of unique reflections | 35,356 (2,943) |
| Refinement statistics | |
| Resolution | 76.0–2.20 |
| $R$work/$R$free (%) | 19.6/23.5 |
| No. of atoms | |
| Protein | 4,302 |
| Solvent | 195 |
| Ligand/ion | 29 |
| Average B factors (Å$^2$) | |
| Protein | 71.8 |
| Solvent | 68.8 |
| Ligand/ion | 76 |
| Wilson B | 48 |
| RMSD | |
| Bond lengths (Å) | 0.002 |
| Bond angles (°) | 0.534 |
| Ramachandran plot | |
| Favoured (%) | 97.3 |
| Allowed (%) | 2.7 |

phosphates (Figs 2B and S2B). R854, the first conserved arginine from this motif, is in a position to interact with the expected lo-cation of the $\gamma$-phosphate in an ATP molecule (Fig 2B). This means of contacting the nucleotide, with potentially both conserved arginine residues, has not been seen in other RecQ family structures to date. A mutational analysis in Bloom syndrome helicase (BLM) indicated that both residues are important for helicase activity with the

performed 7 d after siRNA transfection, and the data are represented relative to non-targeting control siRNA. Data information: In (C), viability data are shown as mean ± SD of three biological repeat experiments. **(D)** Immunofluorescence analysis of $\gamma$-H2AX was performed 72 h after siRNA transfection. The mean nuclear $\gamma$H2AX intensity (a.u.) was quantified after siRNA transfection. Data points shown (n ≥ 120 cells per condition) are derived from a single representative experiment that is consistent with a biological repeat experiment. Scale bar, 20 $\mu$M. **(E)** Mitotic chromosome spread analysis was performed 72 h after siRNA transfection. At the 66 h time-point, cells were treated with 6 h of Nocodazole (1.5 $\mu$M) before spreading to enrich for mitotic stages. As a reference, some chromosome breaks are highlighted by red arrowheads. Each mitotic spread was categorized into less than five breaks or more than five breaks (n ≥ 28 mitotic spreads per condition). Data values and error bars presented here are the mean and the SD, respectively, from biological repeats (n = 2). Scale bar, 10 $\mu$M.
Source data are available for this figure.

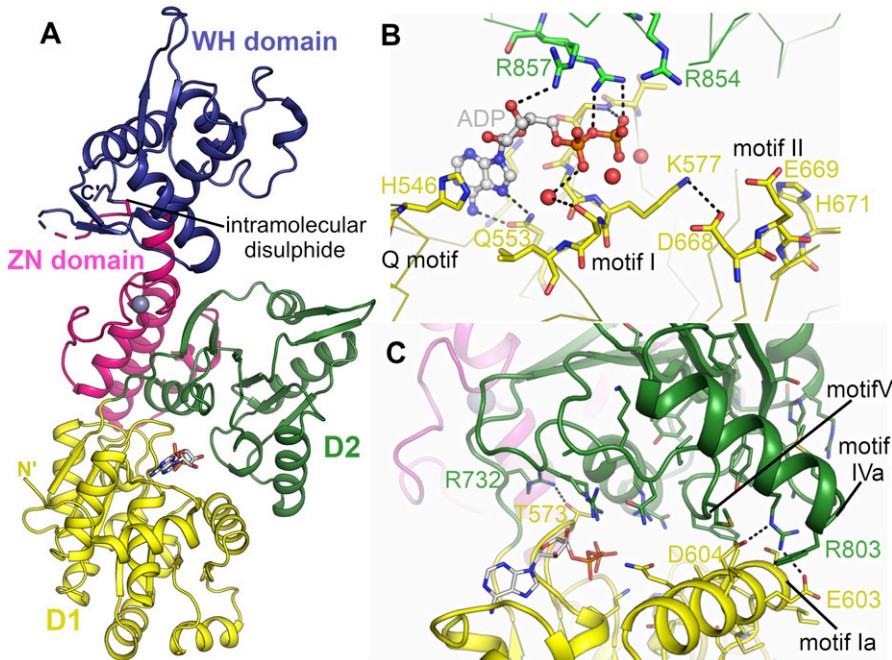

**Figure 2. Structure of WRN helicase catalytic core and the nucleotide-binding site.**
**(A)** Overall structure of WRN helicase catalytic core with domains coloured individually. **(B)** Close-up view of the WRN nucleotide-binding site with conserved helicase motifs and key residues labelled. **(C)** Close-up view of the contact formed and interface between the D1 and D2 domains with key residues and motifs labelled.

equivalent of R857 identified as the transition state stabilizing arginine finger (31).

## Comparisons with previous WRN–WH structures and with other RecQ family members

Structures of the WH domain of WRN have been previously determined in isolation, both with and without DNA (30, 32). Our WH domain structure is in excellent agreement with the previously determined structure in complex with DNA (0.6 Å RMSD) showing only minor variations in the backbone and side chain conformations in the vicinity of the strand separating β-hairpin, presumably as a result of the extensive interactions formed by this region with the DNA junction (30).

The extensive contacts formed between R857 and the ADP nucleotide play a role in defining the relative positioning of the two RecA-like domains, which we and others have found to be variable in RecQ family structures. We have previously performed a systematic analysis of domain positioning in existing RecQ family structures by measuring vector pairs between invariant points on each RecQ domain (33). Using the same analysis on the WRN structure reveals that the positioning of the two RecA domains in WRN is unique (Fig S3), with a significantly more compact arrangement of the two domains, and close contacts formed between motif Ia in D1 and motifs IVa and V in D2 (Fig 2C). On an individual domain basis, the WRN structure is surprisingly most similar to RecQ from *Deinococcus radiodurans* (around 1.6 Å RMSD for both D1 and D2), although the similarity to other human RecQ family members is only slightly lower (generally around 1.8 Å RMSD) (Fig 3A). One unique feature of WRN is an additional β-hairpin inserted between the first helix and second strand of the D2 domain (Figs 3A and S2C). The hairpin is highly reminiscent of the strand separating hairpins found in various helicases and features

a compact type II′ β-turn with a serine (S758) instead of the usual glycine residue at the +1 position. Similar hairpin features have been observed as additions to the second RecA domain in other helicases such as the superfamily I PcrA and superfamily II Hel308, although in the case of WRN the inserted hairpin is on the same face but opposite end of the D2 domain. The amino acid sequence at this region is not well conserved in WRN homologues, although this is also the case for the canonical strand separating hairpin (aa 1,028–1,043) (30) in the WRN–WH domain and in hairpins found in helicases from other organisms.

Another notable difference between our structure of WRN and other RecQ family structures is the position adopted by the WH domain relative to the D2 domain. The positioning of this domain has been found to vary in other RecQ helicase structures, especially in the absence of DNA, whereas in the presence of DNA the complexes are more consistent, with the WH domain packed closely against the helical hairpin of the Zn domain (34, 35). In the WRN helicase core structure, the WH domain is positioned in an opposite orientation and displaced by around 30 Å from the typical WH positioning in RecQ DNA complexes (Fig 3B). The presence of numerous crystal contacts and an intra-molecular disulphide bond (C946 to C1070) indicate that this positioning is not expected to be representative of the DNA bound conformation, although the flexible attachment of this domain may be a feature of the WRN helicase mechanism and alternative conformations may be required for activity on unusual DNA substrates.

## Potential interactions of the WRN HRDC domain with the helicase core

The function of the HRDC domain in various RecQ helicases is currently an active area of research. Early studies on *E. coli* and

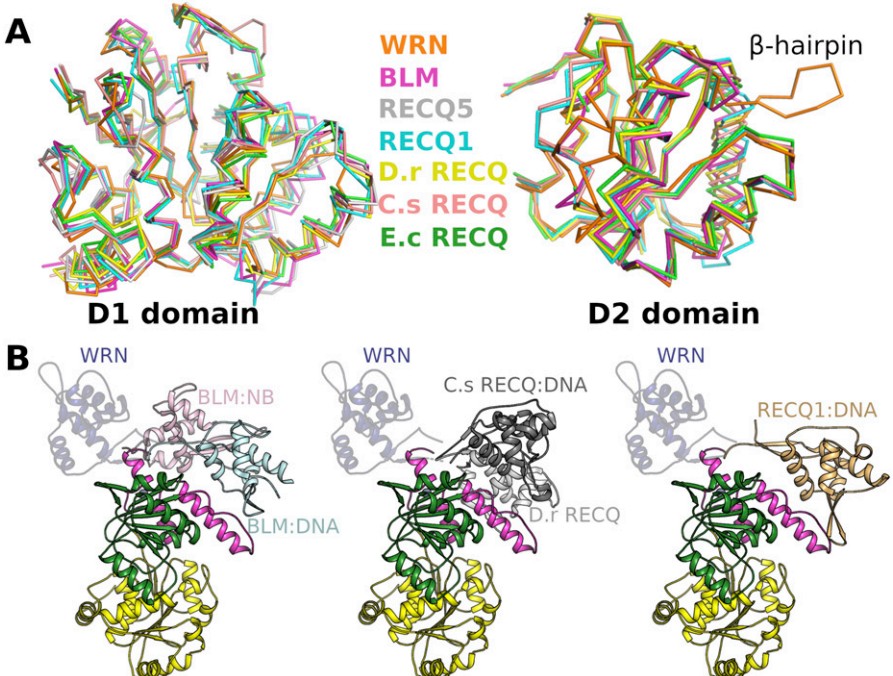

**Figure 3. Comparison of WRN with other members of the RecQ family.**
**(A)** Comparison of current RecQ helicase structures superposed on the basis of the D1 domain (left) and D2 domain (right). **(B)** Comparison of the relative positioning of the winged hHelix domain with respect to the helicase core in various RecQ family structures, alignments were performed on the basis of the D2 domain with BLM-nanobody (NB) complex versus BLM-DNA complex shown on the left, D.r RecQ versus C.s RecQ-DNA complex in the centre and RecQ1–DNA complex shown on the right. The WRN Winged Helix domain is shown throughout in semi-transparent blue.

*Saccharomyces cerevisiae* RecQ proteins indicated a role as an accessory DNA-binding domain, with the isolated HRDC domains having an electropositive surface and displaying binding affinity for single-stranded DNA (ssDNA) in the low to mid micro molar range (36, 37). On the other hand structural studies on isolated HRDC domains from BLM and WRN did not find an electropositive surface and no DNA binding activity could be determined for WRN HRDC, although BLM HRDC was reported to have weak ssDNA-binding affinity (~100 µM) in one study, whereas another failed to detect any binding at all (38, 39, 40). Further clues as to the role of the HRDC domain came from structural investigations of human BLM, which showed that in the presence or absence of DNA the HRDC domain packs tightly against the helicase core and forms interactions with both D1 and D2 domains in a nucleotide dependant manner (34, 41). Subsequent studies with *E. coli* RecQ showed that the HRDC domain suppresses the rate of ATP hydrolysis and DNA unwinding independently of its ability to bind DNA (42). From this it has been suggested that interaction between the helicase core and the HRDC domain is a conserved feature of RecQ helicases, although the effect on the helicase activity may vary according to the roles of the different enzymes.

To investigate this possibility for WRN, we have constructed a model of the possible interface between the HRDC domain and the helicase core using the structure of the WRN HRDC domain determined in isolation together with the WRN helicase core and the relative domain positioning found in the BLM helicase structures (34, 39). In this model, there is generally good shape complementarity between the WRN HRDC domain and its expected interface (Fig 4A), with some minor clashes formed by hydrophobic residues at the C-terminal end of the first helix of the HRDC domain that can be largely relieved by adopting alternative rotamers for the affected residues. The putative interface in the WRN structure is slightly smaller and less polar than in the BLM structure, with significantly fewer salt bridges (1 versus 8) and hydrogen bonds formed (7 versus 17). Nevertheless, the WRN HRDC can be seen to make potentially favourable pairs of interactions to D1 (primarily hydrophobic in nature) and D2 (more polar), with the nucleotide being in close proximity to a number of polar residues in the interface, K1182 and T1180 (Fig 4A). Other unique features of the WRN HRDC domain such as the extended N-terminal helix and C-terminal loop motif (39) are found on the opposite face to the expected interface suggesting an interaction is possible.

We have tested the possibility of interaction between the HRDC domain and the helicase core in solution by NMR using an $^{15}$N-labelled HRDC domain expressed separately. Fig 4B shows an overlay of $^{15}$N-SoFast-HMQC spectra of the HRDC domain. An intensity decrease of several resonances as well as minor chemical shift perturbations can be observed after the addition of WRN (531–950), exhibiting higher solubility than WRN (517–1,093) under the experimental conditions. This effect could only be seen with the addition of high concentrations of WRN ATPase core (250 µM), indicating that the interaction of the unconnected HRDC and helicase domains of WRN is very weak under the experimental conditions. Sequential backbone assignment enabled mapping of the interaction surface onto the structure of the WRN HRDC domain. Fig 4C shows the surface of residues that show a significant intensity decrease ($I_{HRDC+Helicase}/I_{HRDC}$ < 0.4). The mapped interaction surface is in good accordance with the proposed model. The analysis suggests that the HRDC domain interacts with the helicase core and exhibits a similar spatial arrangement as described previously for human BLM. This weak interaction of the unconnected domains may be biologically significant in the context of the full-length protein, in which the WH domain and the HRDC domain are connected by a flexible linker of around 70 residues. This linker

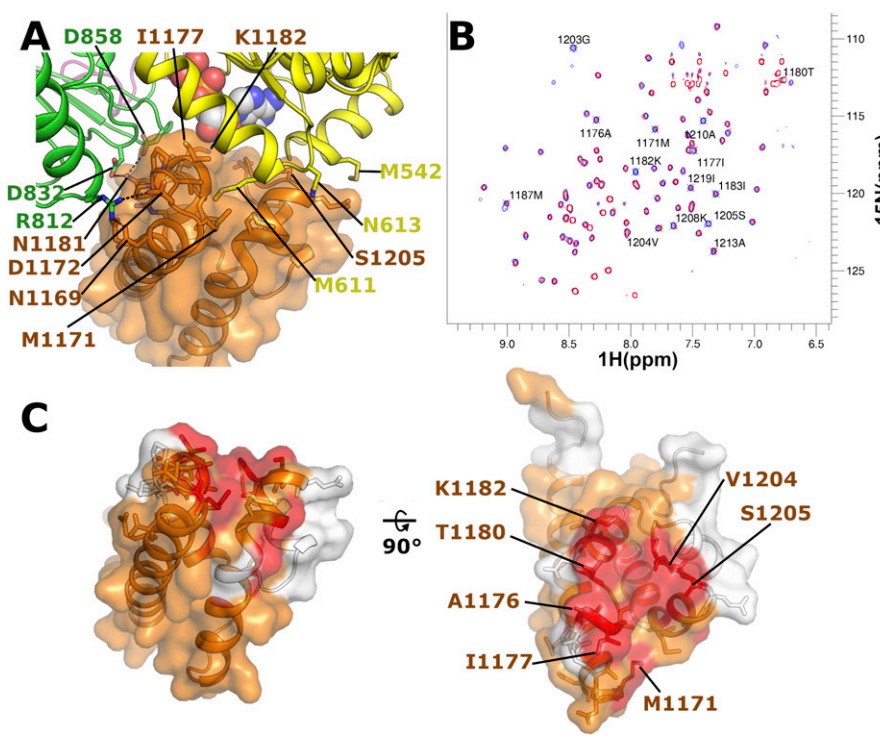

**Figure 4. Examination of possible contacts between the WRN HRDC domain and helicase core.**
**(A)** Structural model of the possible WRN HRDC domain–helicase core interaction interface created by positioning the isolated WRN HRDC structure into its expected position based on the BLM helicase structure. **(B)** Superposition of 2D $^{15}$N SoFast HMQC spectra of 30 $\mu$M $^{15}$N-labelled WRN HRDC in absence (blue) or presence (red) of 250 $\mu$M unlabelled WRN helicase (residues 531–950). Resonances with a strong intensity decrease are highlighted. **(C)** Mapping of the interaction site between the WRN HRDC domain and the helicase core by NMR. Residues whose resonance is strongly affected by the interaction with the helicase domain are highlighted in red. Residues which could not be assigned are highlighted in white. All residues within 4 Å of a helicase core residue in our model are shown in stick format. The left-hand panel shows the HRDC domain in the same orientation as in section A, whereas the right-hand panel shows an orthogonal view with the interface.

is significantly longer than in BLM (around 10 residues), and this additional flexibility may enable the WRN HRDC module to play other roles in addition to forming interactions with the helicase core.

### A model for WRN DNA binding

We have also used the existing available structural information to construct a model for WRN helicase bound to a simple DNA substrate (Fig 5A). The model is constructed by positioning the WRN WH domain DNA complex structure (3AAF) (30) onto the position adopted by the *Cronobacter sakazakii* RecQ–DNA complex. This model was chosen as the WH domain positioning in either the BLM or RecQ1 DNA bound structures, although being broadly similar, give a significant number of steric clashes because of the unusual

conformation adopted by the WRN helical hairpin. The double stranded DNA from the WRN–WH DNA complex structure was extended to include a four nucleotide 3′ overhang. The third and fourth nucleotide of the overhang are in close contact to residues of the conserved helicase motifs IV and V, respectively, a feature common to all RecQ DNA structures determined to date (Fig S4). The first and second nucleotides of the 3′ overhang connect these two DNA elements, with the conformation of these nucleotides being similar to that observed in the RecQ1 DNA structure, although there is significantly more uncertainty over this region as it is quite variable across the various DNA bound RecQ structures determined to date.

In the model, the double-stranded region of the DNA sits in a cleft between the D2 and WH domains and makes extensive interactions with the WH domain in the region of the β-hairpin, as has

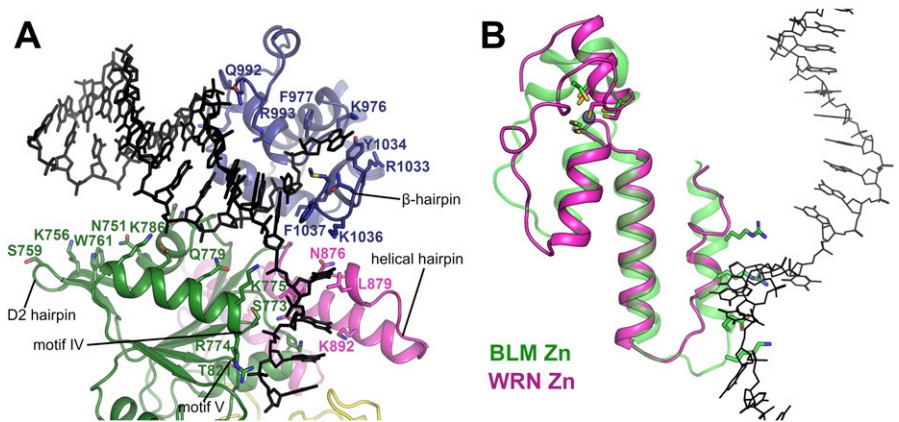

**Figure 5. A model of WRN bound to DNA containing a 3′ overhang.**
**(A)** Overview of the WRN DNA model with predicted DNA contacting residues and motifs labelled. **(B)** Comparison of the zinc-binding domain in BLM (green) and WRN (pink) helicases and its contacts to the 3′ DNA overhang (shown in black stick format), the WRN Zn-binding domain features an extended linker helix, alternate positioning of Zn coordinating residues, and coil conformation of the N-terminal arm of the helical hairpin. Side chain residues from the helical hairpin that form contacts to DNA in the BLM structure are shown in stick format for reference.

been described previously (30). In addition, potential favourable interactions are formed by polar residues of the second helix in D2 such as K775, Q779 and K786. The blunt end of the double stranded DNA is also close to the WRN-specific hairpin insertion on the D2 domain. A number of polar residues from this hairpin and the preceding α-helix point towards the DNA duplex and are suggestive of a possible role in the protein DNA interface, although in the current model, DNA is slightly too distant for formation of direct contacts. The single stranded 3′ overhang passes close to the helical hairpin region of the Zn domain, which in WRN is distinctly different from that found in all other RecQ structures, with the N-terminal helix being predominantly coil rather than helical (Fig 5B). This helix forms significant contacts to the ssDNA in other RecQ structures, generally in the form of hydrophobic contacts from bulky side chain residues contacting the nucleobases. In our WRN DNA model, the equivalent residues are more distant with significantly more room for the DNA to pass unhindered, and additional pockets on the surface created by the helix to coil transition (Fig 5B). One clue as to the possible function of such a structure came from a recent structural study on *C. sakazakii* RecQ in complex with an unwound G-quadruplex DNA, which found a guanine-specific pocket that accommodated a flipped out Guanine nucleobase with residues in this pocket being identified as essential for G4 unwinding (43). The pocket identified for bacterial RecQ is not conserved in WRN, although it may be possible that the pockets formed by the helix coil transition of the helical hairpin, which are in a similar position but on the opposite side of the DNA tract, may play the same role.

### Hydrogen deuterium exchange (HDX) measurements of WRN in solution

We have probed the WRN interaction with both nucleotide and DNA in solution by performing HDX MS measurements of WRN in the presence and absence of both ssDNA and the non-hydrolysable ATP analogue AMP-PNP using a shorter WRN construct (531–950) produced in insect cells lacking the WRN WH domain, which gave excellent peptide coverage of 91% (Table S1). As can be seen in Fig 6A, a significant protection of residues close to the Q motif, and motifs I and III can be seen in the presence of nucleotide. In

particular the peptide spanning residues 571–580 containing motif I shows a reduced deuteration of between 2 and 3 D, which is consistent with the typical mode of nucleotide interaction with motif I, with three consecutive direct hydrogen bonds donated by backbone amides, as found in other RecQ nucleotide structures (Fig S1). With longer incubation periods (4 h) additional protection can be seen for residues from motif VI in the D2 domain (Fig 6A), which is consistent with the extensive interactions between that region and the nucleotide in our crystal structure. Addition of ssDNA alone does not cause any differences in exchange at short labelling times; however, after 4 h of labelling, significant protection can be seen for a peptide containing part of the D2 hairpin and the entire helicase motif IV (Fig 6B), although the sequence coverage for this measurement was significantly lower (77%). Both the D2 hairpin and motif IV are predicted in our WRN DNA complex model to have the potential to interact with DNA, therefore indicating that the HDX data support our modelling studies. These HDX results further suggest that the compact arrangement of D1 and D2 domains found in our crystal structure, and the extensive contacts formed between nucleotide and D2, may be a feature of the WRN protein in solution. The unusual mode of nucleotide interaction with motif I seen in the crystal structure does not appear to be predominant in solution, as revealed by HDX, and thus may be associated with a particular WRN conformational state rather than being a general feature of the WRN protein.

## Discussion

We have determined the crystal structure of WRN helicase core (517–1,093), a highly anticipated structure due to the recently discovered importance of WRN as selective dependency of and therapeutic target in MSI cancer cells. Our structure shows several unique features that may have implications in the WRN helicase mechanism. We show an unusual mode of nucleotide binding with extensive nucleotide interactions formed by residues in the D2 domain that have been confirmed in solution by HDX measurements. These interactions define the relative domain positioning of the D1 and D2 domains, which form a compact arrangement distinct from that seen in other RecQ structures and may represent a

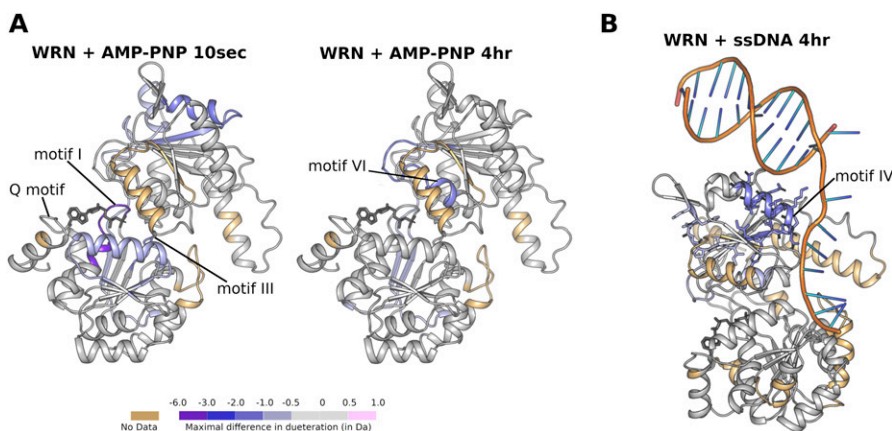

**Figure 6.  Hydrogen deuterium exchange MS measurements of WRN in solution.**
**(A)** Comparative HDX ($D_{AMP-PNP}$–$D_{unbound}$) of WRN in complex with the ATP analogue AMP-PNP mapped onto the WRN structure. Protection can be seen for nucleotide-binding features in D1 (left), whilst protection for nucleotide contacting residues from D2 are observed over longer time scales (right). **(B)** Comparative HDX of WRN ($D_{ssDNA}$–$D_{unbound}$) in complex with single-stranded DNA, mapped on to the WRN DNA-binding model. Protection can be seen for residues in the D2 hairpin and motif IV.

defined state in the WRN catalytic mechanism. Another possible feature of the WRN mechanism is an interaction between the WRN HRDC domain and helicase core. This interaction was demonstrated previously for BLM helicase, were the HRDC association ensures defined conformation of the helicase core via contacts to both D1 and D2. We show via NMR that a similar weak association between the HRDC and helicase core exists for WRN, and mapping of chemical shift perturbations onto the WRN HRDC structure indicates that the interface may be conserved. We have constructed a model for WRN DNA binding, in which the WH domain adopts an alternative position to that observed in the crystal structure. The model suggests possible roles for a WRN-specific insertion in the D2 domain and an unusual helical hairpin in defining the DNA protein interface.

# Materials and Methods

### Cell culture and lentiviral transduction

The human colon cancer cell line HCT 116 was grown in McCoy's 5A medium with GlutaMAX (36600–021; GIBCO) to which 10% FCS was added. Wild-type and mutant codon-optimized, siRNA resistant WRN transgenes containing a C-terminal 3xFLAG tag (designated WRNr) were synthesized and inserted into the lentiviral pLVX-IRES-puro plasmid vector (ClonTech) at GenScript. Lentivirus particles were generated using the Lenti-X Single Shot system (ClonTech) in 293T-Lenti-X cells. HCT 116 cell pools stably transduced with the WRNr transgene carrying lentivirus particles were selected with 2 $\mu$g/ml of Puromycin (P9620; Sigma-Aldrich) added to the normal growth medium. These cell lines were generated using lentiviral transduction. Single cell clones were obtained by limiting dilution. All cell lines used in this study tested negatively for mycoplasma contamination and were further authenticated by short tandem repeat fingerprinting.

### Immunoblotting

Cells were lysed in extraction buffer (50 mM Tris–HCl, pH 8.0, 1% Nonidet P-40, and 150 mM NaCl) to which complete protease inhibitor mix (Roche) and phosphatase inhibitor cocktails (P5726 and P0044; Sigma-Aldrich) were added.

### Antibodies

The antibodies used in this study are as follows: WRN (8H3) mouse mAb (4666, 1/1,000 dilution; Cell Signaling), mouse anti-FLAG (F1804; Sigma-Aldrich, 1/1,000 [immunoblotting] or 1/500 [immunofluorescence] dilution), rabbit anti–phospho-histone H2A.X (Ser139) (2577, 1/800 dilution; Cell Signaling), mouse anti-GAPDH (ab8245, 1/30,000 dilution; Abcam), mouse Alexa Fluor 488 (1/1,000 dilution; Molecular Probes) and secondary rabbit (P0448, 1/1,000 dilution; Dako), mouse anti-IgG-HRP (P0161, 1/1,000 dilution; Dako).

### siRNA transfection and cell viability

For siRNA knock-down experiments, cells were transfected using Lipofectamine RNAiMAX reagent according to the manufacturer's instructions (Invitrogen) supplemented with the following siRNA duplexes: WRN and PLK1 targeting ON-TARGETplus siRNA duplex (J-010378–05, L-003290-00; Dharmacon); ON-TARGETplus Non-targeting Control Pool (D-001810-10; Dharmacon). The final concentration of the siRNA was 20 nM in immunoblotting, immunofluorescence, and chromosome spread experiments. Cell viability experiments were carried out with 10 nM siRNA concentration in 96-well plates with a total volume of 100 $\mu$l per well and a starting number of 1,000 HCT 116 cells per well. Cellular viability was measured 7 d after transfection using CellTiter-Glo reagent (Promega). 100 $\mu$l of the 1:2 diluted CellTiter-Glo solution was directly added to the growth medium, mixed briefly, and incubated for 10 min before the measurement of the luminescence signal.

### Immunofluorescence

Cells were transfected as mentioned above and grown for 72 h. Subsequently the cells were fixed for 15 min using 4% paraformaldehyde, permeabilized for 10 min with 0.2% Triton X-100 in PBS and blocked for 45 min with 3% BSA in PBST (PBS containing 0.01% Triton X-100). Cells were incubated sequentially with primary antibodies that detect either FLAG or phospho-Histone H2A.X (Ser139) and secondary antibodies (Alexa 488; Molecular Probes). Coverslips were mounted, and cells were counterstained on the glass slides using ProLong Gold with DAPI (4′, 6-diamidino-2-phenylindole) (Molecular Probes). Images were collected using an Axio Plan2/AxioCam microscope and image processing was performed with MrC5/Axiovision software (Zeiss). Quantification of $\gamma$-H2AX foci was carried out using segmentation in the Halo software (https://www.indicalab.com/halo/) that identified DAPI-stained nuclei. Subsequently, the corresponding $\gamma$-H2AX mean intensities of the identified nuclei were determined.

### Chromosome spreads

66 h after siRNA transfection, cells were treated with 1.5 $\mu$M nocodazole for 6 h. Cells were swollen in hypotonic buffer for 5 min at room temperature in a solution composed of 40% medium/60% tap water. Fixation was performed three times with freshly made Carnoy's solution (75% methanol and 25% acetic acid). To acquire chromosome spreads, cells in the fixative solution were dropped onto glass slides and air-dried. Slides were later stained with 5% Giemsa (Merck) for 4 min, washed briefly with tap water, and air-dried. The analysis was performed from two independent slides for each condition, and a blind quantification of the chromosome breaks was carried out. Images were acquired using an Axio Plan2/AxioCam microscope and image processing was performed with MrC5/Axiovision software (Zeiss).

### Cloning, overexpression, and purification of WRN helicase domain

WRN constructs corresponding to residues 517–1,093 were cloned in the vector pNIC-CTHF using ligation independent cloning and transformed into *E. coli* LOBSTR cells for overexpression (44). Cells were grown at 37°C in Terrific Broth supplemented with 50 $\mu$g/ml kanamycin until an optical density of 2–3 and induced by the addition of 0.1 mM IPTG and incubated overnight at 18°C. Cells were harvested by centrifugation. For purification, cell pellets were

thawed and resuspended in buffer A (50 mM Hepes, pH 7.5, 500 mM NaCl, 5% glycerol, 30 mM imidazole, and 0.5 mM Tris [2-carboxyethyl] phosphine [TCEP]), with the addition of 1× protease inhibitor set VII (Merck). Cells were lysed by sonication and cell debris pelleted by centrifugation. Lysates were loaded on to a Ni-sepharose gravity flow column (GE Healthcare), washed with 2 column volumes of wash buffer (buffer A supplemented with 45 mM imidazole), and eluted with 300 mM imidazole in buffer A. The purification tag was cleaved with the addition of 1:20 mass ratio of His-tagged TEV protease during overnight dialysis into buffer A. TEV prottease was removed by rebinding to Ni-Sepharose and the flow through and wash fractions were combined, concentrated using a 50,000 mwco centrifugal concentrator and loaded on to size exclusion chromatography using a HiLoad 16/60 Superdex s200 column (GE Healthcare) in buffer A. Fractions containing WRN were pooled, and diluted to 25 mM Hepes, 250 mM NaCl, 2.5% glycerol, 0.25 mM TCEP and loaded onto a 1-ml HiTrap Heparin HP column (GE Healthcare), equilibrated in the same buffer. Proteins were eluted with a 40-ml linear gradient to 50 mM Hepes, 1 M NaCl. Protein concentrations were determined by measurement at 280 nm (NanoDrop) using the calculated molecular mass and extinction coefficients.

WRN (residues 531–950) containing an N-terminal HIS-Zbasic-tag followed by a TEV cleavage site was cloned into a pFB-6HZB transfer vector. Recombinant baculovirus was obtained by transfecting Sf9 cells. Consecutively, Hi5 cells were infected with the virus and cultured in suspension using wave bags at 27°C for 50 h. Cells were harvested and disrupted by sonication in lysis buffer (20 mM Hepes, pH 7.5, 500 mM NaCl, 5% glycerol, 10 mM imidazole, 5 mM $MgCl_2$, 1 mM TCEP, and 1 mM ATP) containing a protease inhibitor cocktail (Complete; Roche). The cell suspension was centrifuged at 27,000$g$ for 40 min. The supernatant was loaded onto a Ni-NTA affinity column (GE Healthcare) equilibrated with lysis buffer and washed until baseline. Beads were treated with 20% elution buffer (lysis buffer plus 250 mM imidazole) to remove unspecific impurities and the protein was eluted with 100% elution buffer. The protein was cleaved by incubation with HIS-TEV protease at 4°C overnight. Afterwards, buffer was exchanged to lysis buffer, using a HiPrep Desalting column (GE Healthcare) and loaded again onto a Ni-NTA affinity column. Untagged Protein still possesses weak affinity to beads, thus a 10% elution buffer step was applied. The combined flow-through and 10% eluate fraction were concentrated by centrifugation, using an Amicon Ultra 30000 MWCO (Merck). The protein was finally loaded onto a HiLoad 16/600 Superdex 200 pg gel-filtration column (GE Healthcare) equilibrated with 50 mM Hepes, pH 7.5, 300 mM NaCl, 5 mM $MgCl_2$, 1 mM TCEP, and 1 mM ATP. The major peak, corresponding to 48 kD in SDS–PAGE analysis, was concentrated to 15 mg/ml.

## Crystallization and structure determination

For crystallization of WRN (517–1,093), the protein peak from the Heparin column was concentrated to 12 mg/ml using a 50,000 MWCO centrifugal concentrator diluted twofold in water (final buffer is 25 mM Hepes, ~150 mM NaCl, and 0.25 mM TCEP) and co crystallized with 5 mM $AlCl_3$, 60 mM NaF, 5 mM ADP, and 5 mM $MgCl_2$ at a final protein concentration of 5.5 mg/ml. WRN crystals appeared between 1 and 2 mo in conditions containing 1 M Na Acetate, 0.1 M cacodylate, pH 6.5. Crystals were cryo-protected by transferring to a solution of mother liquor supplemented with 20% glycerol and flash-cooled in liquid nitrogen. Data were collected at Diamond Light Source beamline I03,

and data were processed with the programs DIALS (45). The structure was solved by molecular replacement using the program PHASER (46) with the RECQL1 (29) and WRN WH structures (30) as starting models. Model building and real space refinement were performed in COOT (47) and the structures refined using PHENIX REFINE (48). A summary of the data collection and refinement statistics is shown in Table 1.

## Expression and purification of $^{15}$N-labelled WRN HRDC domain

The construct for expression of the WRN HRDC domain (1,142–1,242), as described previously (39), was obtained by gene synthesis (GeneArt; Thermo Fisher Scientific) in a donor vector (pDONR-221) and transferred by recombinant cloning into the GST fusion vector pDEST15 (Invitrogen). The plasmid was used to transform *E. coli*, strain BL21 (DE3). An overnight culture in Luria Broth supplemented with 100 µg/ml ampicillin at 37°C was prepared and added to M9 minimal medium supplemented with either $^{15}NH_4Cl$ (0.5 g/l), or $^{15}NH_4Cl$ (0.5 g/l) and [U-$^{13}$C] glucose (2.5 g/l) the next day. At A600 of 0.95, the expression was induced by the addition of 0.25 mM IPTG and incubated at 20°C for 24 h (A600 of 3.1). Cell pellets obtained by centrifugation at 6,000$g$ were stored at –20°C. Cells were solubilized in lysis buffer (20 mM Tris, pH 7.5, 500 mM NaCl, 1 mM TCEP, 5% glycerol) and disrupted by sonication (Sonopuls from Bandelin) on ice. The sonicated lysate was clarified by centrifugation at 27,000$g$ for 40 min. The supernatant was loaded onto a glutathione-Sepharose-4B affinity column (GE Healthcare) equilibrated with lysis buffer and washed until a stable baseline was obtained. The beads were mixed with TEV protease and incubated at 4°C overnight and washed with lysis buffer. The flow-through was concentrated by centrifugation using an Amicon Ultra 10000 MWCO. The concentrated solution was loaded onto a HiLoad 16/600 Superdex 75 pg gel-filtration column (GE Healthcare), equilibrated with 20 mM Tris, pH 7.5, 200 mM NaCl, and 1 mM TCEP. The first major peak (containing TEV) was discarded and the second peak, corresponding to 11 kD in SDS–PAGE analysis, was concentrated to 11.4 mg/ml ($^{15}$N-labelled HRDC) and 13.1 mg/ml ($^{13}C^{15}$N-labelled HRDC).

## Measurement of the interaction between WRN HRDC and WRN helicase

$^1$H-$^{15}$N SoFast HMQC experiments were recorded in 3 mm NMR tubes (200 µl filling) at a protein concentration of $^{15}$N-labelled WRN HRDC of 30 µM ± unlabelled WRN helicase (531–950) 250 µM in sample buffer (Hepes 50 mM, pH 7.5, NaCl 300 mM, ATP 1 mM, $MgSO_4$ 5 mM, TCEP 1 mM, and $D_2O$ 10%). Spectra were recorded on a Bruker Avance III 700 MHz spectrometer equipped with a cryogenically cooled 5 mm TCI probe at a 298 K and 128 scans, 128 f1 increments, and 2,000 data points in f2. Total acquisition time was 1 h.

## NMR assignment of WRN HRDC

3D-NMR spectra for sequential backbone assignment were recorded in a 3 mm NMR tube (180 µl filling) at a protein concentration of $^{13}C^{15}$N-labelled WRN HRDC of 1 mM in sample buffer (Tris 20 mM, pH 7.5, NaCl 200 mM, TCEP 1 mM, and $D_2O$ 10%) on a Bruker Avance III 600 MHz spectrometer equipped with a cryogenically cooled 5 mm TCI probe at a 298 K. The experiments performed included $^1$H-$^{15}$N

SoFast HMQC, 3D best-HNCA, 3D best-HN(CO)CA, 3D best-HNCO, 3D best-HN(CA)CO, 3D best-HNCACB, and 3D best-HN(CO)CACB (49). Spectra were processed with Topspin 3.5 (Bruker BioSpin) and analysed with CcpNMR (50).

## HDX MS measurements of WRN in solution

Deuterium labelling: Deuterium labelling was initiated by diluting 3 $\mu$L of WRN (16.66 $\mu$M; 15 mM Hepes and 150 mM NaCl, pH 7.3) 16-fold in deuterated buffer at room temperature. The labelling reactions were quenched by decreasing the temperature to 0°C and the pH to 2.5 by adding 48 $\mu$L of quench buffer. Quench buffer 1 (100 mM potassium phosphate, pH 2.1) was used for the AMP-PNP experiments and quench buffer 2 (4 M guanidine hydrochloride, 200 mM potassium phosphate, 200 mM sodium chloride, and 100 mM tris [2-carboxyethl] phosphine hydrochloride [TCEP-HCl], pH 2.2) was used for binding experiments where ssDNA was present. Samples were taken at five time points (10 s, 1 min, 10 min, 1 h, and 4 h). The sequence of the ssDNA used in the experiments was as follows: 5′-CCA GGT CGA TAG GTT CGA ATT GGT T-3′. Complexes with AMP-PNP and ssDNA were analysed in a similar way. WRN was incubated with AMP-PNP or ssDNA individually. Mixing ratios protein: AMP-PNP of 1:200 and protein:ssDNA 1:2.5 were used. The protein was allowed to equilibrate with the ligands for 2 min at room temperature before $D_2O$ labelling, which was allowed to proceed from 10 s up to 4 h for each condition.

### Liquid chromatography–mass spectrometry

Upon quenching, the samples were injected immediately into a Waters nanoACQUITY ultra performance liquid chromatography (UPLC) equipped with HDX technology. The samples were digested online using a Waters Enzymate BEH Pepsin Column (2.1 × 30 mm, 5 $\mu$m) at 15°C. The cooling chamber of the UPLC system which housed all the chromatographic elements was held at 0.0°C ± 0.1°C for the entire time of the measurements. Peptides were trapped and desalted on a VanGuard Pre-Column trap (2.1 × 5 mm, ACQUITY UPLC BEH C18, 1.7 $\mu$m [186003975; Waters]) for 3 min at 100 $\mu$l/min. Peptides were then eluted from the trap using a 8–35% gradient of acetonitrile (with 0.1% formic acid) over 8 min at a flow rate of 40 $\mu$l/min and separated using an ACQUITY UPLC C18 HSS T3 1.8 $\mu$m, 1.0 × 50 mm column (186003535; Waters). The back pressure averaged ~7,500 $\psi$ at 0°C and 5% acetonitrile 95% water. The error of determining the deuterium levels was ± 0.15 Da in this experimental setup. To eliminate peptide carryover, a wash solution of (1.5 M guanidinium chloride, 0.8% formic acid, and 4% acetonitrile) was injected over the pepsin column during each analytical run. Mass spectra were acquired using a Waters Synapt G2-Si HDMSE mass spectrometer. The mass spectrometer was calibrated with direct infusion of a solution of glu-fibrinopeptide (F3261; Sigma-Aldrich) at 200 fmol/$\mu$l at a flow rate of 5 $\mu$l/min before data collection. A conventional electrospray source was used, and the instrument was scanned at 0.4 scans/second over the range 50–2,000 m/z with ion mobility engaged. The instrument configuration was the following: capillary voltage 3.2 kV, trap collision energy 4 V, sampling cone 40 V, source temperature 80°C and desolvation temperature 175°C. All comparison experiments were carried out under identical experimental conditions such that deuterium levels were not corrected for back-exchange and are therefore reported as relative.

Peptides were identified using PLGS 3.0.1 (RRID: SCR_016664, 720001408EN; Waters) using three replicates of undeuterated control samples. Raw MS data were imported into DynamX 3.0 (720005145EN; Waters) and filtered as follows: minimum consecutive products: 2; minimum number of products per amino acid: 0.3. Peptides meeting these filtering criteria were further processed automatically by DynamX followed by manual inspection of all processed data. The relative amount of deuterium in each peptide was determined by subtracting the centroid mass of the undeuterated form of the peptide from the deuterated form at each time point and for each condition. These deuterium uptake values were used to generate uptake graphs and difference maps.

# Data Availability

The crystal structure of WRN was deposited in the Protein Databank PDB ID 6YHR. All HDX MS data have been deposited to the ProteomeXchange Consortium via the PRIDE partner repository with the dataset identifier PXD018910.

# Supplementary Information

# Acknowledgements

The Structural Genomics Consortium is a registered charity (number 1097737) that receives funds from AbbVie, Bayer Pharma AG, Boehringer Ingelheim, Canada Foundation for Innovation, Eshelman Institute for Innovation, Genome Canada, Innovative Medicines Initiative (EU/EFPIA) (ULTRA-DD grant no. 115766), Janssen, Merck KGaA Darmstadt Germany, MSD, Novartis Pharma AG, Ontario Ministry of Economic Development and Innovation, Pfizer, São Paulo Research Foundation-FAPESP, Takeda, and Wellcome (106169/ZZ14/Z). MC Ravichandran is a member of the Boehringer Ingelheim Discovery Research global post-doc program. The authors would like to thank Diamond Light Source for beamtime (proposal MX19301) and the staff of beamline I03 for assistance with crystal testing and data collection.

## Author Contributions

JA Newman: conceptualization, investigation, and writing—original draft, review, and editing.
AE Gavard: investigation.
S Lieb: conceptualization and investigation.
MC Ravichandran: conceptualization, investigation, and writing—original draft, review, and editing.
K Hauer: conceptualization and investigation.
P Werni: investigation.
L Geist: investigation.
J Böttcher: investigation.
JR Engen: conceptualization and investigation.
K Rumpel: conceptualization and investigation.

M Samwer: conceptualization, supervision, investigation, and writing—original draft, review, and editing.

M Petronczki: conceptualization, supervision, and writing—original draft, review, and editing.

O Gileadi: conceptualization, supervision, writing—original draft, and project administration.

## Conflict of Interest Statement

The authors declare that they have no conflict of interest.

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
