## [Reviewer comments · Life Science Alliance]

Life Science Alliance

Structure of the helicase core of Werner Helicase, a key target in microsatellite instability cancers

Joseph Newman, Angeline Gavard, Simone Lieb, Madhwesh Ravichandran, Katja Hauer, Patrick Werni, Leonhard Geist, Jark Böttcher, John Engen, Klaus Rumpel, Matthias Samwer, Mark Petronczki, and Opher Gileadi

DOI: <https://doi.org/10.26508/lsa.202000795>

Corresponding authors: Opher Gileadi, University of Oxford

Review Timeline:	Submission Date:	2020-05-26
	Editorial Decision:	2020-06-18
	Revision Received:	2020-10-02
	Editorial Decision:	2020-10-18
	Revision Received:	2020-10-28
	Accepted:	2020-10-28

Scientific Editor: Shachi Bhatt

Transaction Report:

June 18, 2020

Re: Life Science Alliance manuscript #LSA-2020-00795-T

Dr. Opher Gileadi
University of Oxford
Structural Genomics Consortium
Old Road Campus Research Building
Roosevelt Drive
Oxford OX3 7DQ
United Kingdom

Dear Dr. Gileadi,

Thank you for submitting your manuscript entitled "Crystal Structure of Werner Syndrome Helicase, a key target in microsatellite instability cancers" to Life Science Alliance. We have now received reports from the three referees that were asked to evaluate your study, which can be found at the end of this email.

As you will see, all referees think that the findings are of interest, but they also have several comments, concerns and suggestions, indicating that a major revision of the manuscript is necessary to allow publication in LSA. As the reports are below, and we think all points need to be addressed, we will not detail them here. Nevertheless, we think the major task will be to rewrite the manuscript so that it becomes clear what the novel findings in the study are.

Given the constructive referee comments, we would like to invite you to revise your manuscript with the understanding that all referee concerns must be addressed in the revised manuscript and/or in a detailed point-by-point response. Acceptance of your manuscript will depend on a positive outcome of a second round of review. It is LSA's policy to allow a single round of revision only and acceptance of the manuscript will therefore depend on the completeness of your responses included in the next, final version of the manuscript.

Revised manuscripts should be submitted within three months of a request for revision. We are aware that many laboratories cannot function at full efficiency during the current COVID-19/SARS-CoV-2 pandemic and we have therefore extended our 'scooping protection policy' to cover the period required for full revision. Please contact me to discuss the revision should you need additional time, and also if you see a paper with related content published elsewhere.

While you are revising your manuscript, please also attend to the below editorial points to help expedite the publication of your manuscript. Please direct any editorial questions to the journal

office.

Thank you for this interesting contribution to Life Science Alliance. We are looking forward to receiving your revised manuscript.

Sincerely,

Reilly Lorenz
Editorial Office Life Science Alliance
Meyerhofstr. 1
69117 Heidelberg, Germany
t +49 6221 8891 414
e contact@life-science-alliance.org
www.life-science-alliance.org

B. MANUSCRIPT ORGANIZATION AND FORMATTING:

Reviewer #1 (Comments to the Authors (Required)):

Summary:

The authors determined the crystal structure of a WRN helicase domain fragment with ADP bound to a resolution of 2.2 Angstroms. Novel structural features important for nucleotide binding are described. Using NMR, modeling studies performed by the authors suggest an interaction between the noncatalytic C-terminal HRDC domain and the helicase core. In addition to the structural work, the authors performed assays with site-directed WRN mutant -transfected HCT116 colorectal cancer cells that display microsatellite instability (MSI). Similar to previous studies they report that ATP binding/hydrolysis is required for WRN to ensure survival of these cells. WRN exonuclease activity is not required, consistent with previous studies.

Critical Comments:

Title:

- The crystal structure (reported in the current study) is of a WRN ATPase core domain fragment. The title provides the expectation to the reader that the study deals with the "Crystal Structure of Werner Syndrome helicase". This is inaccurate as presented. Secondly, the study has little to do with WRN as a key target in microsatellite instability as there really is no connection of the two experimental efforts, i.e., cell-based work and structural studies. As mentioned below, the cell-based studies largely confirm what was already reported about WRN's role in MSI and its functional requirements by several groups in 2019. The title needs to be rewritten.

Abstract:

- The authors should clearly state in the Abstract the WRN fragment used for the crystal structural studies, including the amino acid boundaries and what domains were included in that fragment and what conserved regions of the protein were absent in the recombinant WRN fragment. This is important as otherwise it is misleading to the readers who might have the impression that a full-length WRN was studied.

Introduction:

- 1st paragraph: While individuals afflicted with Werner syndrome develop more than one type of cancer, there are specific cancer types that are strongly dominant. In contrast, Bloom syndrome truly is a syndrome characterized by a broad range of cancer types. The wording on WS and cancer requires clarification in the text.

- 1st paragraph: It is odd that the authors specifically mention ionizing radiation (IR) as one of the DNA damaging agents that WS patients are sensitive to. Firstly, to my knowledge it is the cells of the patients that were assessed. Secondly, the literature suggests that WS cells are only mildly sensitive to IR, and some of these studies were performed with non-isogenic cell lines. The authors should cite papers that are most relevant for cellular sensitivity to DNA damaging agents or treatments, characterized by carefully controlled experimentation and the greatest differences in drug sensitivity. Interestingly, they cite Refs 3-5 to support the statement, but I am not sure that IR was even used on one of those papers.

- 2nd paragraph: Some of the most relevant cell-based work for WS/WRN deals with its specialized functions at stalled/regressed replication forks performed by the Monnat, Sidorova, and Vindigni labs. This background information in the Introduction is much more biologically significant than some of the in vitro results sporadically mentioned in the Introduction (e.g., base excision repair and WRN, which has been reported by only a limited contingent of researchers and the relevance of the in vitro findings to in vivo roles of WRN is unclear). Overall, the Introduction covers a broad range of topics that are only loosely connected to the focus of the current manuscript.

- 2nd paragraph: "Rewrite Suggestion" needs to be deleted.

- 3rd paragraph: Authors comment about specificity, off-target effects and cytotoxicity of previously discovered WRN helicase inhibitors. While improvements in potency and selectivity may be required for clinical application, these comments are not appropriate here for several reasons: 1) Cited Ref. 20 examined viability at a single timepoint 7 days after initial drug exposure which may very well have missed the sensitivity window and half-life of drug bioactivity in tissue culture, as performed by Brosh lab studies. 2) Cited Ref. 26 did not address NSC19630 drug effect on WRN bioactivity; 3) The current study did not address effects of published WRN inhibitors (NSC19630 or NSC617145 in vitro or in vivo).

Results:

- Figure 1: It is unclear why these data which largely recapitulate several papers published in 2019 are even included in this manuscript which is focused on structural analysis of a WRN helicase domain fragment. The WRN mutants characterized by cell-based studies focus on WRN helicase or exonuclease activity, both of which were already published for their roles in MSI in cancer cell lines. If the authors were to attempt to draw a novel connection between their structural work and the cell-based work, they might have addressed the biological importance of the HRDC domain for MSI as they report a novel interaction of the WRN HRDC with the helicase core from the modeling analysis. As the construction of the manuscript currently stands, I see little value in the cell-based work as it does not significantly extend our understanding of WRN's role in MSI from what was previously published by multiple groups in 2019.

- The connectivity of the studies involving the WRN ATPase core fragment crystal structure and the NMR/modeling analysis of the HRDC domain is lacking. The former lacks the HRDC domain and the latter lacks the ATPase core. The disconnect here is apparent and presents a deficiency in experimental design and overall robustness of interpretation of results in a wholesome manner.

- WRN HRDC domain: I find it odd that the authors did not discuss in more depth the architecture of the WRN HRDC study based on their modeling studies compared to what is known about the structural/biochemical/biological features of the conserved HRDC domain in BLM. The HRDC domain in BLM has been studied more extensively than that of WRN in these contexts, and the potential importance of BLM's HRDC domain in nucleic acid metabolism (e.g., conformational changes in BLM; importance for BLM/topo double HJ dissolution (Hickson lab)).

Summary:

- The Summary should provide greater insight to the importance of the current study for understanding from a structure-function perspective what was really learned in terms of WRN's nucleic acid interaction properties and catalytic mechanism of action. The Results/Summary should emphasize what significantly new and important insights are gleaned from the current study of WRN domain fragments compared to what was already known about conserved RecQ domain architecture.

Reviewer #2 (Comments to the Authors (Required)):

Newman et al examine the importance of the enzymatic domains of the Werner helicase in a cancer cell line and determine the structure of helicase core of Werner. The authors find that mutations of the Werner ATPase residues mimic depletion of native WRN in cellular responses. Furthermore the structure reveals several novel features of the Werner helicase core. Additional NMR and HDX experiments are used to probe specific questions regarding domain-domain and DNA-Werner interfaces. The structural work is quite exciting and reports an important new structure that many have anticipated for decades. Several issues (listed below) remain to be addressed in the manuscript prior to its acceptance.

Major issues:

- (1) More explanation of the results in Figures 1E (right) and S1 are needed. It is not clear to a non-specialist what I am looking at in the cell images.
- (2) Representative electron density would be useful. In particular, it would be great to see an Fo-Fc map for Arg857 (which is modelled as two rotamers).
- (3) I don't understand the conclusion of the HRDC HSQC experiment. No interaction is observed between the HRDC domain and the helicase core. The authors state that "This indicates that potential interactions of the HRDC with the helicase core are significantly less extensive than that found for BLM". Has the same experiment has been performed with BLM? It seems the conclusion is that, if there is an interaction between the HRDC and helicase core domains of Werner, it is likely weak and the domains must be tethered together for the interaction to occur. Without the identical experiment being performed with BLM, a comparison seems premature.

Minor issues:

- (1) First paragraph, page 2. The authors have left an editorial statement in place ("Rewrite suggestion:"). Please be certain that all comments from the writing process are removed from the manuscript.
- (2) Last line, page 2. There is an extra comma in the line.
- (3) Table S1 has changes tracked during manuscript preparation.
- (4) In the crystallization and structure determination methods section (page 9), the authors state that protein was diluted 2-fold in water prior to crystallization. What were the buffer conditions prior to dilution? This information may be important for others who would like to reproduce the crystals.
- (5) The molecular replacement methods (page 9) differ from the description in results (page 3). The methods section states that RecQ1 was used as the search model whereas the results section states that BLM and the WRN WH domain were used as search models. Please clarify this point.

Reviewer #3 (Comments to the Authors (Required)):

Newman et al. report a crystal structure of WRN fragment 528-1072 (ATPase domain with C-terminal WH domain) in DNA-free form, with additional experiments supporting importance of the ATPase activity in the cells.

Many structures of RECQ-family members, including those of WRN, are available today both in DNA-free and DNA-bound forms, but the study is interesting because it adds the first structural view of the ATPase domain of WRN. Statistics of X-ray structure determination are of good quality and the structure may contribute to the inhibitor development of WRN.

However, the problem in this paper is that the authors do not at all refer to the previous structures of WRN 956-1064 (WH domain; references [1-2]), which is a famous DNA-binding module of WRN and also a major part of the structure observed in this study.

[1] "Solution structure of a multifunctional DNA- and protein-binding motif of human Werner syndrome protein." Hu et al., Proc Natl Acad Sci USA. 2005, 102(51), 18379-84.

[2] "Structural basis for DNA strand separation by the unconventional winged-helix domain of RecQ helicase WRN." Kitano et al., Structure. 2010, 18(2), 177-87.

Title

"Crystal Structure of Werner Syndrome Helicase, a key target in microsatellite instability cancers"
The title is overstating because it usually means "Structure of full-length WRN" (or structure of a large fragment containing all structured domains). WRN is a 1432 amino acid protein that is composed of more than four domains (Figure 1A) while the present structure includes only two of them (545 amino acids). I recommend modifying the title so that it more specifically describe the determined structure; for example "Crystal structure of catalytic core of Werner helicase, a key target ..." or "Structure of Werner helicase catalytic core ...". The same attention should be paid throughout the main text and in Figure 2 legend.

Page 2, line 11-12 from the bottom

"determine the first crystal structure of the catalytic core of WRN helicase ..." -> "determine the first crystal structure of the "full" catalytic core of WRN helicase ..."

The authors determine the structure of WRN 528-1072, a fragment that includes ATPase domain (528-955) and WH domain (956-1064). WH domain is known to be attached to the C-terminus of ATPase domain by a flexible linker. As described above, 3D structures of WRN WH were determined more than 10 years ago; the first group determined an NMR structure of WRN WH in DNA-free form [1] and the second determined a crystal structure of WRN WH in complex with DNA, showing that WH directly binds dsDNA and catalyzes DNA-strand separation using the beta-hairpin motif [2]. The authors should refer to these structures and indicate that the WH structure in the study is also the same (or not).

Page 6, lines 16-27, and Figure 5A

"The model is constructed by positioning the WRN WH domain onto the position adopted by the *Chronobacter sakazakii* RecQ-DNA complex."

It is unclear which structure of WRN WH domain was used in the construction of this DNA-binding model since no reference for the WRN WH-DNA complex structure [2] is given. It is known that structural changes are induced in WRN WH when binding to DNA. Therefore, structure of WRN WH in DNA-bound form should be used rather than that without DNA; the model should be built by overlaying and positioning WRN WH-DNA complex (PDB ID 3AAF) onto the position adopted by the *Chronobacter sakazakii* RecQ-DNA complex so that both the proteins and DNA backbones fit closely.

Page 1, lines 1-6 from the bottom

How did the authors learn these domain boundaries so accurately? References should be given.
"a Zinc binding subdomain (residues 869-994)" -> "a Zinc binding subdomain (residues 869-"955")".

Page 2, line 13

"Rewrite Suggestion:"

What does this mean?

Page 4, line 22

"the catalytically essential K577 does not form direct hydrogen bonds to the β -phosphate,"
Mutation K577M has been believed to prevent ATP-binding (page 2, a line at the bottom). Is the structure compatible with this idea or not?

Page 5, lines 8-10

"the canonical strand separating hairpin (aa 1028-1043) in the WRN WH domain and in hairpins found in helicases from other organisms"

References that identified the canonical strand separating hairpin are missing.

Page 5, lines 17-18

"an intra-molecular disulphide bond (C946 to C1070) indicate that this positioning is not expected to be representative of the DNA bound conformation"

The C946-C1070 disulphide bond should be visualized in Figure 2A because the authors propose that this disulphide bond is a crystallization artifact that results in the inappropriate position of WH domain.

Page 5, line 6 from the bottom

"its expected interface (Figure 2B)" -> "its expected interface (Figure "4A")"?

Page 6, lines 8-13

"This indicates that potential interactions of the HRDC with the helicase core are significantly less extensive than that found for BLM."

Figure 4A should be moved to Supplementary figures because the authors themselves conclude that this structural model is quite unlikely in solution.

Figure 1 legend

"essential for essential for" -> "essential for"

Figure 2 legend

"Structure of WRN helicase" -> "Structure of WRN helicase catalytic core"

Figure 3A right panel

It is difficult to understand the beta-hairpin structure with this backbone model. A new supplementary figure, a close-up view of the hairpin in the stick-model with key residues labeled, would be helpful.

Figure 3B

Please use the consistent names through the figure and text.

"BLM nanobody complex vs BLM DNA" -> "BLM-nanobody (NB) complex and BLM-DNA complex"

"Bacterial RecQ APO vs Bacterial RecQ DNA" -> "D.r RECQ and C.s RECQ-DNA complex"

"RecQ1" -> "RECQ1-DNA complex"

Figure S1 legend

"transgenic cell lines cell lines" -> "transgenic cell lines"?

Response to reviewers' comments

We wish to thank all three reviewers for their careful consideration of our manuscript and their constructive comments. We have taken this opportunity, prompted by the reviewers' comments to perform additional experiments on the nature of the interaction between the WRN HRDC domain and helicase core. These experiments and the other changes detailed below we believe have fully addressed the reviewers concerns and collectively these changes significantly strengthen the manuscript.

Reviewer #1 (Comments to the Authors (Required)):

Summary:

The authors determined the crystal structure of a WRN helicase domain fragment with ADP bound to a resolution of 2.2 Angstroms. Novel structural features important for nucleotide binding are described. Using NMR, modeling studies performed by the authors suggest an interaction between the noncatalytic C-terminal HRDC domain and the helicase core. In addition to the structural work, the authors performed assays with site-directed WRN mutant -transfected HCT116 colorectal cancer cells that display microsatellite instability (MSI). Similar to previous studies they report that ATP binding/hydrolysis is required for WRN to ensure survival of these cells. WRN exonuclease activity is not required, consistent with previous studies.

Critical Comments:

Title:

- The crystal structure (reported in the current study) is of a WRN ATPase core domain fragment. The title provides the expectation to the reader that the study deals with the "Crystal Structure of Werner Syndrome helicase". This is inaccurate as presented. Secondly, the study has little to do with WRN as a key target in microsatellite instability as there really is no connection of the two experimental efforts, i.e., cell-based work and structural studies. As mentioned below, the cell-based studies largely confirm what was already reported about WRN's role in MSI and its functional requirements by several groups in 2019. The title needs to be rewritten.

We accept the reviewer's point (and that of reviewer 3) about the title and have changed it in the revised version to more accurately represent our work, the title now reads

"Crystal structure of the helicase core of Werner Syndrome helicase, a key target in microsatellite instability cancers"

We agree with the reviewer's comment that the cell-based work is largely a continuation of our work in our 2019 papers. While we showed that WRN containing an ATP-binding mutation (K577M) is unable to rescue viability and an accumulation of γ -H2AX foci upon endogenous WRN depletion in our previous report, it remained unknown if ATP-hydrolysis is critical for WRN function in MSI-H cells. Hence the cell-based work used the Walker B mutation (E669A) to show that the loss ATP turnover by WRN helicase results in severe genome instability in MSI-H cells (observed by accumulation of γ -H2AX foci and chromosome breaks). The other 2019 papers do not highlight this point and hence it was shown for the first time in our manuscript. Since the ATP-hydrolysis is an indispensable function of WRN helicase in MSI-H cells inhibition of ATP turnover would be a prime therapeutic strategy to attack MSI-H cancers and further highlights the need for a crystal structure of the ATPase core. Our manuscript goes on to

address this concern by providing the first crystal structure of the WRN helicase ATPase core which would inform future attempts to develop WRN helicase inhibitors. After careful consideration of the reviewer's comments certain sections of the results section from the cell-based work were adapted to better convey the above points.

Abstract:

- The authors should clearly state in the Abstract the WRN fragment used for the crystal structural studies, including the amino acid boundaries and what domains were included in that fragment and what conserved regions of the protein were absent in the recombinant WRN fragment. This is important as otherwise it is misleading to the readers who might have the impression that a full-length WRN was studied.

Again we are happy to change the abstract to include the regions that our structure covers. We have altered the 4th sentence of the abstract, which now reads.

“We further present the crystal structure of an ADP bound form of the WRN helicase core (517-1093) at 2.2 Å resolution. The crystal structure covers the two helicase subdomains and the winged helix domain but not the C-terminal HRDC domain or N-terminal exonuclease domain. “

Introduction:

- 1st paragraph: While individuals afflicted with Werner syndrome develop more than one type of cancer, there are specific cancer types that are strongly dominant. In contrast, Bloom syndrome truly is a syndrome characterized by a broad range of cancer types. The wording on WS and cancer requires clarification in the text.

We agree with the reviewer and are happy to clarify this point further in the text, the following has been added to the 1st paragraph.

“and an increased risk of development of cancers, specifically thyroid cancer, melanoma, soft tissue sarcoma and osteosarcoma.”

- 1st paragraph: It is odd that the authors specifically mention ionizing radiation (IR) as one of the DNA damaging agents that WS patients are sensitive to. Firstly, to my knowledge it is the cells of the patients that were assessed. Secondly, the literature suggests that WS cells are only mildly sensitive to IR, and some of these studies were performed with non-isogenic cell lines. The authors should cite papers that are most relevant for cellular sensitivity to DNA damaging agents or treatments, characterized by carefully controlled experimentation and the greatest differences in drug sensitivity. Interestingly, they cite Refs 3-5 to support the statement, but I am not sure that IR was even used on one of those papers.

We did mention that these sensitivities were exhibited by WS cells, and the papers in references 3-5 were cited for their insights into the general sensitivities and aberrations exhibited by WS cells (mentioned previously in the sentence) and not specifically for the effects of ionizing radiation. We can see how the reviewer may have got this impression and have altered this part to define more carefully what each citation covers and have removed the statement about ionizing radiation, since the sensitivity reported is mild and other studies may not have shown significant effects.

“On a cellular level, cells cultured from WS patients exhibit slow growth (3), chromosome aberrations (4), genome instability and an increased frequency of telomere shortening and loss(4), and are sensitive to various DNA damaging agents that induce inter-strand crosslinks (5).”

- 2nd paragraph: Some of the most relevant cell-base work for WS/WRN deals with its specialized functions at stalled/regressed replication forks performed by the Monnat, Sidorova, and Vindigni labs. This background information in the Introduction is much more biologically significant than some of the in vitro results sporadically mentioned in the Introduction (e.g., base excision repair and WRN, which has been reported by only a limited contingent of researchers and the relevance of the in vitro findings to in vivo roles of WRN is unclear). Overall, the Introduction covers a broad range of topics that are only loosely connected to the focus of the current manuscript.

The reviewer may well be correct that specialized function on stalled or regressed replication forks are more relevant than its role in base excision repair (although the authors of those studies may not agree). We are not really trying to comment on the relevance of these studies to in vivo roles of WRN, our main point in this paragraph is to demonstrate the wide range of pathways that WRN has been linked via a diverse set of protein protein interactions (without missing out any major pathway) to illustrate that WRN likely plays several roles in the cell. We have adapted the point about base excision repair and added a sentence to the 2nd paragraph of the introduction to further highlight the role of WRN in stalled replication forks.

“WRN has also been implicated in base excision repair through an interaction with DNA polymerase beta, stimulating its strand displacement DNA synthesis activity (12).”

And in the same paragraph we have added.

“Specialized functions for WRN have been found in promoting replication fork progression after DNA damage or replication fork arrest (12) (13), with WRN appearing to act in concert the DNA2 nuclease where it promotes degradation of reversed fork structures (14).”

- 2nd paragraph: "Rewrite Suggestion" needs to be deleted.

This has been removed

- 3rd paragraph: Authors comment about specificity, off-target effects and cytotoxicity of previously discovered WRN helicase inhibitors. While improvements in potency and selectivity may be required for clinical application, these comments are not appropriate here for several reasons: 1) Cited Ref. 20 examined viability at a single timepoint 7 days after initial drug exposure which may very well have missed the sensitivity window and half-life of drug bioactivity in tissue culture, as performed by Brosh lab studies. 2) Cited Ref. 26 did not address NSC19630 drug effect on WRN bioactivity; 3) The current study did not address effects of published WRN inhibitors (NSC19630 or NSC617145 in vitro or in vivo).

We thank the reviewer for highlighting this, the respective paragraph was adapted. Indeed the validation and optimization of the reported compounds can significantly profit from the structural enablement. NSC19630 (EX00078634) exhibits in our hand a weak inhibition in a WRN enzymatic assay

(IC50 28µM) and leads to a significant destabilization of WRN protein in a thermal shift experiment. (-7.5°). NSC617145 (EX00008163) exhibits in our hand strong inhibition in a WRN enzymatic assay (IC50 0.5µM) without affecting the thermal stability. We removed the respective statements and highlighted the importance of elucidation of the mode of action, which deserves further investigation.

Results:

- Figure 1: It is unclear why these data which largely recapitulate several papers published in 2019 are even included in this manuscript which is focused on structural analysis of a WRN helicase domain fragment. The WRN mutants characterized by cell-based studies focus on WRN helicase or exonuclease activity, both of which were already published for their roles in MSI in cancer cell lines. If the authors were to attempt to draw a novel connection between their structural work and the cell-based work, they might have addressed the biological importance of the HRDC domain for MSI as they report a novel interaction of the WRN HRDC with the helicase core from the modeling analysis. As the construction of the manuscript currently stands, I see little value in the cell-based work as it does not significantly extend our understanding of WRN's role in MSI from what was previously published by multiple groups in 2019.

This comment has been addressed above. Please refer to the point 1 in the Critical comments section.

- The connectivity of the studies involving the WRN ATPase core fragment crystal structure and the NMR/modeling analysis of the HRDC domain is lacking. The former lacks the HRDC domain and the latter lacks the ATPase core. The disconnect here is apparent and presents a deficiency in experimental design and overall robustness of interpretation of results in a wholesome manner.

- WRN HRDC domain: I find it odd that the authors did not discuss in more depth the architecture of the WRN HRDC study based on their modeling studies compared to what is known about the structural/biochemical/biological features of the conserved HRDC domain in BLM. The HRDC domain in BLM has been studied more extensively than that of WRN in these contexts, and the potential importance of BLM's HRDC domain in nucleic acid metabolism (e.g., conformational changes in BLM; importance for BLM/topo double HJ dissolution (Hickson lab)).

We don't really see a big disconnect here. The question of the role of the HRDC domain is relevant to the entire RecQ family and whilst we would have been delighted to determine the structure of WRN helicase and HRDC domains together we are limited to what we are able to crystallize. To investigate the function of WRN HRDC domain we considered the possibility of a conserved interaction between the HRDC and helicase core as found in the structures of BLM. We accept that the conclusion of our modelling and NMR studies were not definitive, and since the highest concentration used in our original NMR studies (20 µM) was below the apparent Kd in the BLM study (30-100 µM) we could not make any firm statements on the existence of this interaction in WRN. To address this concern we have tested the interaction between the HRDC domain and the WRN helicase again by NMR (¹⁵N-labeled HRDC) using higher concentrations of the WRN helicase construct. We could confirm weak interaction between the two domains and map the interaction surface onto the structure of the HRDC-domain using NMR backbone

assignment. Importantly the residues displaying chemical shifts in the HRDC domain were in good agreement with the predicted interface in our model supporting the view that HRDC core interactions are a feature of both BLM and WRN. The respective part was adapted and rewritten.

Summary:

- The Summary should provide greater insight to the importance of the current study for understanding from a structure-function perspective what was really learned in terms of WRN's nucleic acid interaction properties and catalytic mechanism of action. The Results/Summary should emphasize what significantly new and important insights are gleaned from the current study of WRN domain fragments compared to what was already known about conserved RecQ domain architecture.

We have adapted the summary to place more emphasis on the new insights into the WRN mechanism and DNA binding. The summary paragraph now reads.

“Our structure shows several unique features that may have implications in the WRN helicase mechanism. We show an unusual mode of nucleotide binding with extensive nucleotide interactions formed by residues in the D2 domain that have been confirmed in solution by HDX measurements. These interactions define the relative domain positioning of the D1 and D2 domains which form a compact arrangement distinct from that seen in other RecQ structures and may represent a defined state in the WRN catalytic mechanism. Another possible feature of the WRN mechanism is an interaction between the WRN HRDC domain and helicase core, this interaction was demonstrated previously for BLM helicase where the HRDC association ensures defined conformation of the helicase core via contacts to both D1 and D2. We show via NMR that a similar weak association between the HRDC and helicase core exists for WRN, and mapping of chemical shift perturbations on to the WRN HRDC structure indicates that the interface may be conserved. We have constructed a model for WRN DNA binding in which the WH domain adopts an alternative position to that observed in the crystal structure. The model suggests possible roles for a WRN specific insertion in the D2 domain and an unusual helical hairpin in defining the DNA protein interface.”

Reviewer #2 (Comments to the Authors (Required)):

Newman et al examine the importance of the enzymatic domains of the Werner helicase in a cancer cell line and determine the structure of helicase core of Werner. The authors find that mutations of the Werner ATPase residues mimic depletion of native WRN in cellular responses. Furthermore the structure reveals several novel features of the Werner helicase core. Additional NMR and HDX experiments are used to probe specific questions regarding domain-domain and DNA-Werner interfaces. The structural work is quite exciting and reports an important new structure that many have anticipated for decades. Several issues (listed below) remain to be addressed in the manuscript prior to its acceptance.

Major issues:

(1) More explanation of the results in Figures 1E (right) and S1 are needed. It is not clear to a non-specialist what I am looking at in the cell images.

We agree with the reviewer and have added red arrowheads to help guide the non-specialist readers to observe the chromosome breaks with ease. We thank the reviewer for their suggestion.

(2) Representative electron density would be useful. In particular, it would be great to see an Fo-Fc map for Arg857 (which is modelled as two rotamers).

We are happy to include some electron density images. We have added a new panel to the supplementary figure S2 showing both the final refined 2fo-fc map of the active site and a Fo-Fc omit maps in which the dual conformation of Arg857 has been removed from the model for map calculation.

(3) I don't understand the conclusion of the HRDC HSQC experiment. No interaction is observed between the HRDC domain and the helicase core. The authors state that "This indicates that potential interactions of the HRDC with the helicase core are significantly less extensive than that found for BLM". Has the same experiment has been performed with BLM? It seems the conclusion is that, if there is an interaction between the HRDC and helicase core domains of Werner, it is likely weak and the domains must be tethered together for the interaction to occur. Without the identical experiment being performed with BLM, a comparison seems premature.

We accept the reviewer's point (also mentioned by Reviewer 1 above) our original NMR experiment was performed with a relatively low concentration of the WRN helicase core and we accept that our conclusion was not really clear. We have substantially expanded the experimental work covering the interaction between the HRDC domain and the helicase core, showing that there is a weak interaction between the two domains (see also above), as is the case for a similar study performed with BLM helicase this weak association is probably biologically significant given the tethering of the two domains in the full length protein in vivo. We have adapted this section of the manuscript and also our conclusions based on this new data.

Minor issues:

(1) First paragraph, page 2. The authors have left an editorial statement in place ("Rewrite suggestion:"). Please be certain that all comments from the writing process are removed from the manuscript.

This has been removed.

(2) Last line, page 2. There is an extra comma in the line.

This has been removed

(3) Table S1 has changes tracked during manuscript preparation.

This has been corrected.

(4) In the crystallization and structure determination methods section (page 9), the authors state that protein was diluted 2-fold in water prior to crystallization. What were the buffer conditions prior to dilution? This information may be important for others who would like to reproduce the crystals.

The protein buffer is not precisely defined as the peak fractions from a gradient elution of a heparin column were pooled as the final purification step. We did clarify this point in the text and also indicated an approximate buffer composition based on the conductivity trace.

(5) The molecular replacement methods (page 9) differ from the description in results (page 3). The methods section states that RecQ1 was used as the search model whereas the results section states that BLM and the WRN WH domain were used as search models. Please clarify this point.

The molecular replacement model used was RecQ1 for the helicase core and the WRN WH domain (pdb IDs 2WWY and 3AAF). We have updated both sections accordingly.

Reviewer #3 (Comments to the Authors (Required)):

Newman et al. report a crystal structure of WRN fragment 528-1072 (ATPase domain with C-terminal WH domain) in DNA-free form, with additional experiments supporting importance of the ATPase activity in the cells.

Many structures of RECQ-family members, including those of WRN, are available today both in DNA-free and DNA-bound forms, but the study is interesting because it adds the first structural view of the ATPase domain of WRN. Statistics of X-ray structure determination are of good quality and the structure may contribute to the inhibitor development of WRN.

However, the problem in this paper is that the authors do not at all refer to the previous structures of WRN 956-1064 (WH domain; references [1-2]), which is a famous DNA-binding module of WRN and also a major part of the structure observed in this study.

[1] "Solution structure of a multifunctional DNA- and protein-binding motif of human Werner syndrome protein." Hu et al., Proc Natl Acad Sci USA. 2005, 102(51), 18379-84.

[2] "Structural basis for DNA strand separation by the unconventional winged-helix domain of RecQ helicase WRN." Kitano et al., Structure. 2010, 18(2), 177-87.

We apologize for omitting these references, this was a mistake on our part and not intentional. As discussed below we did use the structure of the WRN WH in complex with DNA to construct our WRN DNA model and we mentioned several features of that structure as being important in WRN DNA binding. We have modified our manuscript to include these citations.

<Major points>

Title

"Crystal Structure of Werner Syndrome Helicase, a key target in microsatellite instability cancers"

The title is overstating because it usually means "Structure of full-length WRN" (or structure of a large fragment containing all structured domains). WRN is a 1432 amino acid protein that is composed of more than four domains (Figure 1A) while the present structure includes only two of them (545 amino acids). I recommend modifying the title so that it more specifically describe

the determined structure; for example "Crystal structure of catalytic core of Werner helicase, a key target ..." or "Structure of Werner helicase catalytic core ...". The same attention should be paid throughout the main text and in Figure 2 legend.

We accept this point, which was also raised by reviewer 1 and we have adapted our title and text appropriately.

Page 2, line 11-12 from the bottom

"determine the first crystal structure of the catalytic core of WRN helicase ..." -> "determine the first crystal structure of the "full" catalytic core of WRN helicase ..."

The authors determine the structure of WRN 528-1072, a fragment that includes ATPase domain (528-955) and WH domain (956-1064). WH domain is known to be attached to the C-terminus of ATPase domain by a flexible linker. As described above, 3D structures of WRN WH were determined more than 10 years ago; the first group determined an NMR structure of WRN WH in DNA-free form [1] and the second determined a crystal structure of WRN WH in complex with DNA, showing that WH directly binds dsDNA and catalyzes DNA-strand separation using the beta-hairpin motif [2]. The authors should refer to these structures and indicate that the WH structure in the study is also the same (or not).

As discussed above we have included these citations and have also added a small section to the results where we compare our WH structure with those determined previously in isolation. The following has been added to the 2nd paragraph under the subheading "Comparisons with other RecQ family members".

"Structures of the WH domain of WRN have been previously determined in isolation both with and without DNA (34,42) with our WH domain structure being in excellent agreement with the previously determined DNA complex (0.6 Å RMSD) showing only minor variations in backbone and sidechain conformations in the vicinity of the strand separating β -hairpin presumably as a result of the extensive interactions formed by this region with the DNA junction (34)."

Page 6, lines 16-27, and Figure 5A

"The model is constructed by positioning the WRN WH domain onto the position adopted by the *Chronobacter sakazakii* RecQ-DNA complex."

It is unclear which structure of WRN WH domain was used in the construction of this DNA-binding model since no reference for the WRN WH-DNA complex structure [2] is given. It is known that structural changes are induced in WRN WH when binding to DNA. Therefore, structure of WRN WH in DNA-bound form should be used rather than that without DNA; the model should be built by overlaying and positioning WRN WH-DNA complex (PDB ID 3AAF) onto the position adopted by the *Chronobacter sakazakii* RecQ-DNA complex so that both the proteins and DNA backbones fit closely.

Our model was indeed constructed as the reviewer has suggested by positioning 3AAF structure into the expected WH position as found in the *Chronobacter sakazakii* RecQ-DNA complex. We have updated the text to make this more clear and added a reference to the WH DNA complex.

<Minor points>

Page 1, lines 1-6 from the bottom

How did the authors learn these domain boundaries so accurately? References should be given. "a Zinc binding subdomain (residues 869-994)" -> "a Zinc binding subdomain (residues 869-955)".

We have defined domain boundaries based on our own visual examination of the structures, we agree that this may be somewhat subjective as to the exact cut off between one domain and another but as the authors of the first study to describe the structure of this region we feel it appropriate to do so. Our assessment of the domain boundaries appears to be broadly in line with what has been described in other RecQ family members and in some other WRN studies.

Page 2, line 13

"Rewrite Suggestion:"
What does this mean?

This was added in error and has now been removed.

Page 4, line 22

"the catalytically essential K577 does not form direct hydrogen bonds to the β -phosphate," Mutation K577M has been believed to prevent ATP-binding (page 2, a line at the bottom). Is the structure compatible with this idea or not?

This is a good point, we would speculate that whilst the particular conformation we crystallized does not feature extensive interactions between K577 and the nucleotide, these interactions would likely still play important roles in the wider catalytic cycle. We have added this point to the results section under the subheading "Structure of the nucleotide binding site".

"Somewhat unexpectedly the catalytically essential K577 does not form direct hydrogen bonds to the β -phosphate, instead forming polar contacts with motif II (Walker B motif). This is in contrast to what has been observed for other RecQ family member structures, and we expect this residue to still play an important role in WRN ATP binding, perhaps forming the contact in other conformational states."

Page 5, lines 8-10

"the canonical strand separating hairpin (aa 1028-1043) in the WRN WH domain and in hairpins found in helicases from other organisms"
References that identified the canonical strand separating hairpin are missing.

We have added this reference

Page 5, lines 17-18

"an intra-molecular disulphide bond (C946 to C1070) indicate that this positioning is not expected to be representative of the DNA bound conformation"
The C946-C1070 disulphide bond should be visualized in Figure 2A because the authors propose that this disulphide bond is an crystallization artifact that results in the inappropriate position of WH domain.

This is a good suggestion, we have altered Figure 2A to now show these residues in stick format with a slight change of angle required to reveal these details in addition to the other relevant features.

Page 5, line 6 from the bottom

"its expected interface (Figure 2B)" -> "its expected interface (Figure "4A")"

This has been changed

Page 6, lines 8-13

"This indicates that potential interactions of the HRDC with the helicase core are significantly less extensive than that found for BLM."

Figure 4A should be moved to Supplementary figures because the authors themselves conclude that this structural model is quite unlikely in solution.

As detailed above in response to comments made by reviewer 1 and 2, we have repeated the NMR binding studies at higher concentrations and find a low affinity interaction with chemical shift perturbations that match the expected interface in our model. Thus in the light of these new findings our model becomes more informative and we believe it is appropriate to include in the main manuscript.

Figure 1 legend

"essential for essential for" -> "essential for"

This has been changed

Figure 2 legend

"Structure of WRN helicase" -> "Structure of WRN helicase catalytic core"

This has been changed

Figure 3A right panel

It is difficult to understand the beta-hairpin structure with this backbone model. A new supplementary figure, a close-up view of the hairpin in the stick-model with key residues labeled, would be helpful.

We agree that in figure 3A it is hard to see the details of the hairpin, although the purpose of the figure was to show that this feature is significantly different in WRN that in other RecQ structures. We do show more details of this hairpin in the WRN DNA model in figure 5A where the hairpin is highlighted and selected residues are shown in stick format. We are happy to include a further supplementary figure (Figure S1c) that shows this in more detail.

Figure 3B

Please use the consistent names through the figure and text.

"BLM nanobody complex vs BLM DNA" -> "BLM-nanobody (NB) complex and BLM-DNA complex"

"Bacterial RecQ APO vs Bacterial RecQ DNA" -> "D.r RECQ and C.s RECQ-DNA complex"
"RecQ1" -> "RECQ1-DNA complex"

We have updated the nomenclature to be more consistent throughout.

Figure S1 legend

"transgenic cell lines cell lines" -> "transgenic cell lines"?

This has been changed

October 18, 2020

RE: Life Science Alliance Manuscript #LSA-2020-00795-TR

Dr. Opher Gileadi
University of Oxford
Structural Genomics Consortium
Old Road Campus Research Building
Roosevelt Drive
Oxford OX3 7DQ
United Kingdom

Dear Dr. Gileadi,

Thank you for submitting your revised manuscript entitled "Structure of the helicase core of Werner Helicase, a key target in microsatellite instability cancers". We would be happy to publish your paper in Life Science Alliance pending some minor text edits requested by the reviewers and final revisions necessary to meet our formatting guidelines.

Along with the requests below, please also attend to the following:

- please separate the results and discussion sections into 2 separate sections - a Results section and a Discussion section
- please upload your supplementary figures as single files
- please use the [10 author names, et al.] format in your references (i.e. limit the author names to the first 10)
- please add your supplementary figure legends to the main manuscript text (directly under your main figure legends)
- please double check your figure callouts and add callouts for Figure S1B&C; you have figure callouts for S1D and S1E (these are missing from legend and figure)
- please provide the source data for Fig 1B
- when submitted the final revised manuscript, please specify the category for this manuscript

A. FINAL FILES:

B. MANUSCRIPT ORGANIZATION AND FORMATTING:

Sincerely,

Shachi Bhatt, Ph.D.

Executive Editor
Life Science Alliance
<https://www.life-science-alliance.org/>
Tweet @SciBhatt @LSAJournal

Reviewer #1 (Comments to the Authors (Required)):

First sentence of Summary needs to be revised as it is inaccurate.
The sentence should be revised as follows: "We have determined the crystal structure of the WRN helicase core (517-1093)."

As the sentence currently reads, the authors make it sound like they have determined the crystal structure of full-length WRN, which is not true.

Reviewer #2 (Comments to the Authors (Required)):

The revised version is now ready for publication.

Reviewer #3 (Comments to the Authors (Required)):

The authors have done a lot in response to my review comments, but there are still several points that need to be addressed before publication of the paper.

Page 5, lines 11-25

- > As discussed above we have included these citations and
- > have also added a small section to the results where we
- > compare our WH structure with those determined previously
- > in isolation. The following has been added to the 2nd
- > paragraph under the subheading
- > "Comparisons with other RecQ family members".

The previously-determined two WRN WH structures should not be included in the latter half of "Comparisons with other RecQ family members" because the two structures are not "other" at all but "identical" to the structure determined in the study. Such inappropriate writing will make all readers being confused and misunderstand what the novel findings in the study are. Therefore, the authors should move the 2nd paragraph with a new subheading "Comparisons with the previous WRN WH structures" before the section "Comparisons with other RecQ family members" with the beginning of the sentence "Another notable difference" -> "A notable difference".

Page 2, line 10 from the bottom

"determine the first crystal structure of the catalytic core of WRN helicase ..." -> "determine the first crystal structure of the full catalytic core of WRN helicase ...", "determine the first crystal structure of the ATPase core of WRN helicase ..." or "determine the crystal structure of the catalytic core of WRN helicase ...".

As previously written in my first review, the sentence is inaccurate because the structure is not fully

new nor first. The authors should avoid overstating but should make it clear what the novel findings in the study are. In fact, their WH structure, located in an inappropriate position due to the artificial disulphide-bond formation, is only the third one.

Page 7, line 6 from the bottom

"the crystal structure of WRN helicase" -> "the crystal structure of WRN helicase catalytic core"

Page 6, line 8

"the extended N-terminal helix and C-terminal loop motif (39)"

The reference number (39) should not be deleted but is needed here.

Page 10, line 10 from the bottom

"as described previously ," -> "as described previously (39),"

The reference number (39) should not be deleted but is needed here.

October 28, 2020

RE: Life Science Alliance Manuscript #LSA-2020-00795-TRR

Dr. Opher Gileadi
University of Oxford
Centre for Medicines Discovery
Old Road Campus Research Building
Roosevelt Drive
Oxford OX3 7DQ
United Kingdom

Dear Dr. Gileadi,

Thank you for submitting your Research Article entitled "Structure of the helicase core of Werner Helicase, a key target in microsatellite instability cancers". It is a pleasure to let you know that your manuscript is now accepted for publication in Life Science Alliance. Congratulations on this interesting work.

*****IMPORTANT:** If you will be unreachable at any time, please provide us with the email address of an alternate author. Failure to respond to routine queries may lead to unavoidable delays in publication.*******

DISTRIBUTION OF MATERIALS:

Again, congratulations on a very nice paper. I hope you found the review process to be constructive and are pleased with how the manuscript was handled editorially. We look forward to future exciting

submissions from your lab.

Sincerely,

Shachi Bhatt, Ph.D.

Executive Editor

Life Science Alliance

<https://www.lsjournal.org/>
